# GENERATIVE PRINCIPAL COMPONENT ANALYSIS

**Zhaoqiang Liu** *
National University of Singapore
dcslizha@nus.edu.sg

**Jiulong Liu** *
Chinese Academy of Sciences
jiulongliu@lsec.cc.ac.cn

**Subhroshekhar Ghosh**
National University of Singapore
subhrowork@gmail.com

**Jun Han**
PCG, Tencent
junhanjh@tencent.com

**Jonathan Scarlett**
National University of Singapore
scarlett@comp.nus.edu.sg

## ABSTRACT

In this paper, we study the problem of principal component analysis with generative modeling assumptions, adopting a general model for the observed matrix that encompasses notable special cases, including spiked matrix recovery and phase retrieval. The key assumption is that the first principal eigenvector lies near the range of an $L$-Lipschitz continuous generative model with bounded $k$-dimensional inputs. We propose a quadratic estimator, and show that it enjoys a statistical rate of order $\sqrt{\frac{k \log L}{m}}$, where $m$ is the number of samples. Moreover, we provide a variant of the classic power method, which projects the calculated data onto the range of the generative model during each iteration. We show that under suitable conditions, this method converges exponentially fast to a point achieving the above-mentioned statistical rate. This rate is conjectured in (Aubin et al., 2019; Cocola et al., 2020) to be the best possible even when we only restrict to the special case of spiked matrix models. We perform experiments on various image datasets for spiked matrix and phase retrieval models, and illustrate performance gains of our method to the classic power method and the truncated power method devised for sparse principal component analysis.

## 1 INTRODUCTION

Principal component analysis (PCA) is one of the most popular techniques for data processing and dimensionality reduction (Jolliffe, 1986), with an abundance of applications such as image recognition (Hancock et al., 1996), gene expression data analysis (Alter et al., 2000), and clustering (Ding & He, 2004; Liu & Tan, 2019). PCA seeks to find the directions that capture maximal variances in vector-valued data. In more detail, letting $\mathbf{x}_1, \mathbf{x}_2, \ldots, \mathbf{x}_m$ be $m$ realizations of a random vector $\mathbf{x} \in \mathbb{R}^n$ with a population covariance matrix $\bar{\boldsymbol{\Sigma}} \in \mathbb{R}^{n \times n}$, PCA aims to reconstruct the top principal eigenvectors of $\bar{\boldsymbol{\Sigma}}$. The first principal eigenvector can be computed as follows:

$$\mathbf{u}_1 = \arg \max_{\mathbf{w} \in \mathbb{R}^n} \mathbf{w}^T \boldsymbol{\Sigma} \mathbf{w} \quad \text{s.t.} \quad \|\mathbf{w}\|_2 = 1, \tag{1}$$

where the empirical covariance matrix is defined as $\boldsymbol{\Sigma} := \frac{1}{m} \sum_{i=1}^m (\mathbf{x}_i - \mathbf{c})(\mathbf{x}_i - \mathbf{c})^T$, with $\mathbf{c} := \frac{1}{m} \sum_{i=1}^m \mathbf{x}_i$. In addition, subsequent principal eigenvectors can be estimated by similar optimization problems subject to being orthogonal to the previous vectors.

PCA is consistent in the conventional setting where the dimension of the data $n$ is relatively small compared to the sample size $m$ (Anderson, 1962), but leads to rather poor estimates in the high-dimensional setting where $m \ll n$. In particular, it has been shown in various papers that the empirical principal eigenvectors are no longer consistent estimates of their population counterparts (Nadler,

---

*Corresponding authors.

2008; Johnstone & Lu, 2009; Jung & Marron, 2009; Birnbaum et al., 2013). In order to tackle the curse of dimensionality, a natural approach is to impose certain structural constraints on the principal eigenvectors. A common assumption is that the principal eigenvectors are sparse, and this gives rise to the problem of sparse principal component analysis (SPCA) (Zou et al., 2006). In particular, for recovering the top principal eigenvector, the optimization problem of SPCA is given by

$$\mathbf{u}_1 = \arg \max_{\mathbf{w} \in \mathbb{R}^n} \mathbf{w}^T \mathbf{\Sigma} \mathbf{w} \quad \text{s.t.} \quad \|\mathbf{w}\|_2 = 1, \|\mathbf{w}\|_0 \leq s, \tag{2}$$

where $\|\mathbf{w}\|_0 = |\{i \,:\, w_i \neq 0\}|$ denotes the number of non-zero entries of $\mathbf{w}$, and $s \in \mathbb{N}$ represents the sparsity level. In addition to reducing the effective number of parameters, the sparsity assumption also enhances the interpretability (Zou et al., 2006).

Departing momentarily from the PCA problem, recent years have seen tremendous advances in deep generative models in a wide variety of real-world applications (Foster, 2019). This has motivated a new perspective of the related problem of compressed sensing (CS), in which the standard sparsity assumption is replaced by a generative modeling assumption. That is, the underlying signal is assumed to lie near the range of a (deep) generative model (Bora et al., 2017). The authors of (Bora et al., 2017) characterized the number of samples required to attain an accurate reconstruction, and also presented numerical results on image datasets showing that compared to sparsity-based methods, generative priors can lead to large reductions (e.g., a factor of 5 to 10) in the number of measurements needed to recover the signal up to a given accuracy. Additional numerical and theoretical results concerning inverse problems using generative models have been provided in (Van Veen et al., 2018; Dhar et al., 2018; Heckel & Hand, 2019; Jalal et al., 2020; Liu & Scarlett, 2020a; Ongie et al., 2020; Whang et al., 2020; Jalal et al., 2021; Nguyen et al., 2021), among others.

In this paper, following the developments in both PCA/SPCA and inverse problems with generative priors, we study the use of generative priors in principal component analysis (GPCA), which gives a generative counterpart of SPCA in (2), formulated as follows:

$$\mathbf{u}_1 = \arg \max_{\mathbf{w} \in \mathbb{R}^n} \mathbf{w}^T \mathbf{\Sigma} \mathbf{w} \quad \text{s.t.} \quad \mathbf{w} \in \text{Range}(G), \tag{3}$$

where $G$ is a (pre-trained) generative model, which we assume has a range contained in the unit sphere of $\mathbb{R}^n$.[1] Similarly to SPCA, the motivation for this problem is to incorporate prior knowledge on the vector being recovered (or alternatively, a prior preference), and to permit meaningful recovery and theoretical bounds even in the high-dimensional regime $m \ll n$.

## 1.1 RELATED WORK

In this subsection, we summarize some relevant works, which can roughly be divided into (i) the SPCA problem, and (ii) signal recovery with generative models.

**SPCA**: It has been proved that the solution of the SPCA problem in (2) attains the optimal statistical rate $\sqrt{s \log n / m}$ (Vu & Lei, 2012), where $m$ is the number of samples, $n$ is the ambient dimension, and $s$ is the sparsity level of the first principal eigenvector. However, due to the combinatorial constraint, the computation of (2) is intractable. To address this computational issue, in recent years, an extensive body of practical approaches for estimating sparse principal eigenvectors have been proposed in the literature, including (d'Aspremont et al., 2007; Vu et al., 2013; Chang et al., 2016; Moghaddam et al., 2006; d'Aspremont et al., 2008; Jolliffe et al., 2003; Zou et al., 2006; Shen & Huang, 2008; Journée et al., 2010; Hein & Bühler, 2010; Kuleshov, 2013; Yuan & Zhang, 2013; Asteris et al., 2011; Papailiopoulos et al., 2013), just to name a few.

Notably, statistical guarantees for several approaches have been provided. The authors of (Yuan & Zhang, 2013) propose the truncated power method (TPower), which adds a truncation operation to the power method to ensure the desired level of sparsity. It is shown that this approach attains the optimal statistical rate under appropriate initialization. Most approaches for SPCA only focus on estimating the first principal eigenvector, with a certain deflation method (Mackey, 2008) being leveraged to reconstruct the rest. However, there are some exceptions; for instance, an iterative thresholding approach is proposed in (Ma, 2013), and is shown to attain a near-optimal statistical rate

---

[1]Similarly to (Liu et al., 2020; 2021a), we assume that the range of $G$ is contained in the unit sphere for convenience. Our results readily transfer to general (unnormalized) generative models by considering its normalized version. See Remark 1 for a detailed discussion.

when estimating multiple individual principal eigenvectors. In addition, the authors of (Cai et al., 2013) propose a regression-type method that gives an optimal principal subspace estimator. Both works (Ma, 2013; Cai et al., 2013) rely on the assumption of a spiked covariance model to ensure a good initial vector. To avoid the spiked covariance model assumption, the work (Wang et al., 2014) proposes a two-stage procedure that attains the optimal subspace estimator in polynomial time.

**Signal recovery with generative models**: Since the seminal work (Bora et al., 2017), there has been a substantial volume of papers studying various inverse problems with generative priors. One of the problems more closely related to PCA is spectral initialization in phase retrieval, which amounts to solving an eigenvalue problem. Phase retrieval with generative priors has been studied in (Hand et al., 2018; Hyder et al., 2019; Jagatap & Hegde, 2019; Wei et al., 2019; Shamshad & Ahmed, 2020; Aubin et al., 2020; Liu et al., 2021b). In particular, the work (Hand et al., 2018) models the underlying signal as being in the range of a fully-connected ReLU neural network with no offsets, and all the weight matrices of the ReLU neural network are assumed to have i.i.d. zero-mean Gaussian entries. In addition, the neural network needs to be sufficiently expansive in the sense that $n_i \geq \Omega(n_{i-1} \log n_{i-1})$, where $n_i$ is the width of the $i$-th layer. Under these assumptions, the authors establish favorable global optimization landscapes for the corresponding objective, and derive a near-optimal sample complexity upper bound. They minimize the objective function directly over the latent space in $\mathbb{R}^k$ using gradient descent, which may suffer from local minima in general optimization landscapes (Hyder et al., 2019; Shah & Hegde, 2018).

In (Aubin et al., 2020), the assumptions on the neural network are similar to those in (Hand et al., 2018), relaxing to general activation functions (beyond ReLU) and $n_i \geq \Omega(n_{i-1})$. The authors focus on the high dimensional regime where $n, m, k \to \infty$ with the ratio $m/n$ being fixed, and assume that the input vector in $\mathbb{R}^k$ is drawn from a separable distribution. They derive sharp asymptotics for the information-theoretically optimal performance and for the associated approximate message passing (AMP) algorithm. Both works (Hand et al., 2018; Aubin et al., 2020) focus on noiseless phase retrieval. When only making the much milder assumption that the generative model is Lipschitz continuous, with no assumption on expansiveness, Gaussianity, and offsets, a spectral initialization step (similar to that of sparse phase retrieval) is typically required in order to accurately reconstruct the signal (Netrapalli et al., 2015; Candès et al., 2015). The authors of (Liu et al., 2021b) propose an optimization problem similar to (3) for the spectral initialization for phase retrieval with generative models. It was left open in (Liu et al., 2021b) how to solve (or approximate sufficiently accurately) the optimization problem in practice.

Understanding the eigenvalues of spiked random matrix models has been a central problem of random matrix theory, and spiked matrices have been widely used in the statistical analysis of SPCA. Recently, theoretical guarantees concerning spiked matrix models with generative priors have been provided in (Aubin et al., 2019; Cocola et al., 2020). In particular, in (Aubin et al., 2019), the assumptions are similar to those in (Aubin et al., 2020), except that the neural network is assumed to have exactly one hidden layer. The Bayes-optimal performance is analyzed, and it is shown that the AMP algorithm can attain this optimal performance. In addition, the authors of (Aubin et al., 2019) propose the linearized approximate message passing (LAMP) algorithm, which is a spectral algorithm specifically designed for single-layer feedforward neural networks with no bias terms. The authors show its superiority to classical PCA via numerical results on the Fashion-MNIST dataset. In (Cocola et al., 2020), the same assumptions are made as those in (Hand et al., 2018) on the neural network, and the authors demonstrate the benign global geometry for a nonlinear least squares objective. Similarly to (Hand et al., 2018), the objective is minimized over $\mathbb{R}^k$ using a gradient descent algorithm, which can get stuck in local minima for general global geometries.

## 1.2 CONTRIBUTIONS

The main contributions of this paper are as follows:

- We study eigenvalue problems with generative priors (including GPCA), and characterize the statistical rate of a quadratic estimator similar to (3) under suitable assumptions.

- We propose a variant of the classic power method, which uses an additional projection operation to ensure that the output of each iteration lies in the range of a generative model. We refer to our method as projected power method (PPower). We further show that under appropriate conditions (most notably, assuming exact projections are possible), PPower

obtains a solution achieving a statistical rate that is conjectured to be optimal in (Aubin et al., 2019; Cocola et al., 2020) for spiked matrix models.

- For the spiked matrix and phase retrieval models, we perform numerical experiments on image datasets, and demonstrate that when the number of samples is relatively small compared to the ambient dimension, PPower leads to significantly better performance compared to the classic power method and TPower.

Compared to the above-mentioned works that use generative models, we make no assumption on expansiveness, Gaussianity, and offsets for the generative model, and we consider a data model that simultaneously encompasses both spiked matrix and phase retrieval models, among others.

## 1.3 NOTATION

We use upper and lower case boldface letters to denote matrices and vectors respectively. We write $[N] = \{1, 2, \cdots, N\}$ for a positive integer $N$, and we use $\mathbf{I}_N$ to denote the identity matrix in $\mathbb{R}^{N \times N}$. A *generative model* is a function $G : \mathcal{D} \to \mathbb{R}^n$, with latent dimension $k$, ambient dimension $n$, and input domain $\mathcal{D} \subseteq \mathbb{R}^k$. We focus on the setting where $k \ll n$. For a set $S \subseteq \mathbb{R}^k$ and a generative model $G : \mathbb{R}^k \to \mathbb{R}^n$, we write $G(S) = \{G(\mathbf{z}) : \mathbf{z} \in S\}$. We use $\|\mathbf{X}\|_{2 \to 2}$ to denote the spectral norm of a matrix $\mathbf{X}$. We define the $\ell_q$-ball $B_q^k(r) := \{\mathbf{z} \in \mathbb{R}^k : \|\mathbf{z}\|_q \le r\}$ for $q \in [0, +\infty]$. $\mathcal{S}^{n-1} := \{\mathbf{x} \in \mathbb{R}^n : \|\mathbf{x}\|_2 = 1\}$ represents the unit sphere in $\mathbb{R}^n$. The symbols $C, C', C''$ are absolute constants whose values may differ from line to line.

## 2 PROBLEM SETUP

In this section, we formally introduce the problem, and overview some important assumptions that we adopt. Except where stated otherwise, we will focus on the following setting:

- We have a matrix $\mathbf{V} \in \mathbb{R}^{n \times n}$ satisfying

$$\mathbf{V} = \bar{\mathbf{V}} + \mathbf{E}, \tag{4}$$

where $\mathbf{E}$ is a perturbation matrix, and $\bar{\mathbf{V}}$ is assumed to be positive semidefinite (PSD). For PCA and its constrained variants, $\mathbf{V}$ and $\bar{\mathbf{V}}$ can be thought of as the empirical and population covariance matrices, respectively.

- We have an $L$-Lipschitz continuous generative model $G : B_2^k(r) \to \mathbb{R}^n$. For convenience, similarly to that in (Liu et al., 2020), we assume that $\text{Range}(G) \subseteq \mathcal{S}^{n-1}$.

  **Remark 1.** *For a general (unnormalized) $L$-Lipschitz continuous generative model $G$, we can instead consider a corresponding normalized generative model $\tilde{G} : \mathcal{D} \to \mathcal{S}^{n-1}$ as in (Liu et al., 2021b), where $\mathcal{D} := \{\mathbf{z} \in B_2^k(r) : \|G(\mathbf{z})\|_2 > R_{\min}\}$ for some $R_{\min} > 0$, and $\tilde{G}(\mathbf{z}) = \frac{G(\mathbf{z})}{\|G(\mathbf{z})\|_2}$. Then, the Lipschitz constant of $\tilde{G}$ becomes $L/R_{\min}$. For a $d$-layer neural network, we typically have $L = n^{\Theta(d)}$ (Bora et al., 2017). Thus, we can set $R_{\min}$ to be as small as $1/n^{\Theta(d)}$ without changing the scaling laws, which makes the dependence on $R_{\min}$ very mild.*

- We aim to solve the following eigenvalue problem with a generative prior:[2]

$$\hat{\mathbf{v}} := \max_{\mathbf{w} \in \mathbb{R}^n} \mathbf{w}^T \mathbf{V} \mathbf{w} \quad \text{s.t.} \quad \mathbf{w} \in \text{Range}(G). \tag{5}$$

Note that since $\text{Range}(G) \subseteq \mathcal{S}^{n-1}$, we do not need to impose the constraint $\|\mathbf{w}\|_2 = 1$. Since $\mathbf{V}$ is not restricted to being an empirical covariance matrix, (5) is more general than GPCA in (3). However, we slightly abuse terminology and also refer to (5) as GPCA.

- To approximately solve (5), we use a projected power method (PPower), which is described by the following iterative procedure:[3]

$$\mathbf{w}^{(t+1)} = \mathcal{P}_G(\mathbf{V}\mathbf{w}^{(t)}), \tag{6}$$

---

[2]To find the top $r$ rather than top one principal eigenvectors that are in the range of a generative model, we may follow the common approach to use the iterative deflation method for PCA/SPCA: Subsequent principal eigenvectors are derived by recursively removing the contribution of the principal eigenvectors that are calculated already under the generative model constraint. See for example (Mackey, 2008).

[3]In similar iterative procedures, some works have proposed to replace $\mathbf{V}$ by $\mathbf{V} + \rho \mathbf{I}_n$ for some $\rho \in \mathbb{R}$ to improve convergence, e.g., see Deshpande et al. (2014).

---

**Algorithm 1** A projected power method for GPCA (`PPower`)

---

**Input**: $\mathbf{V}$, number of iterations $T$, pre-trained generative model $G$, initial vector $\mathbf{w}^{(0)}$
**Procedure:** Iterate $\mathbf{w}^{(t+1)} = \mathcal{P}_G(\mathbf{V}\mathbf{w}^{(t)})$ for $t \in \{0, 1, \dots, T-1\}$, and return $\mathbf{w}^{(T)}$

---

where $\mathcal{P}_G(\cdot)$ is the projection function onto $G(B_2^k(r))$,[4] and the initialization vector $\mathbf{w}^{(0)}$ may be chosen either manually or randomly, e.g., uniform over $\mathcal{S}^{n-1}$. Often the initialization vector $\mathbf{w}^{(0)}$ plays an important role and we may need a careful design for it. For example, for phase retrieval with generative models, as mentioned in (Liu et al., 2021b, Section V), we may choose the column corresponding to the largest diagonal entry of $\mathbf{V}$ as the starting point. See also (Yuan & Zhang, 2013, Section 3) for a discussion on the initialization strategy for TPower devised for SPCA. We present the algorithm corresponding to (6) in Algorithm 1.

**Remark 2.** *To tackle generalized eigenvalue problems encountered in some specific applications, there are variants of the projected power method, which combine a certain power iteration with additional operations to ensure sparsity or enforce other constraints. These applications include but not limited to sparse PCA (Journée et al., 2010; Yuan & Zhang, 2013), phase synchronization (Boumal, 2016; Liu et al., 2017), the hidden clique problem (Deshpande & Montanari, 2015), the joint alignment problem (Chen & Candès, 2018), and cone-constrained PCA (Deshpande et al., 2014; Yi & Neykov, 2020). For example, under the simple spiked Wigner model (Perry et al., 2018) for the observed data matrix $\mathbf{V}$ with the underlying signal being assumed to lie in a convex cone, the authors of (Deshpande et al., 2014) show that cone-constrained PCA can be computed efficiently via a generalized projected power method. In general, the range of a Lipschitz-continuous generative model is non-convex and not a cone. In addition, we consider a matrix model that is more general than the spiked Wigner model.*

Although it is not needed for our main results, we first state a lemma (proved in Appendix A) that establishes a monotonicity property with minimal assumptions, only requiring that $\mathbf{V}$ is PSD; see also Proposition 3 of Yuan & Zhang (2013) for an analog in sparse PCA. By comparison, our main results in Section 4 will make more assumptions, but will also provide stronger guarantees. Note that the PSD assumption holds, for example, when $\mathbf{E} = \mathbf{0}$, or when $\mathbf{V}$ is a sample covariance matrix.

**Lemma 1.** *For any $\mathbf{x} \in \mathbb{R}^n$, let $Q(\mathbf{x}) = \mathbf{x}^T \mathbf{V}\mathbf{x}$. Then, if $\mathbf{V}$ is PSD, the sequence $\{Q(\mathbf{w}^{(t)})\}_{t>0}$ for $\mathbf{w}^{(t)}$ in (6) is monotonically non-decreasing.*

## 3 SPECIALIZED DATA MODELS AND EXAMPLES

In this section, we make more specific assumptions on $\mathbf{V} = \bar{\mathbf{V}} + \mathbf{E}$, starting with the following.

**Assumption 1** (Assumption on $\bar{\mathbf{V}}$). *Assume that $\bar{\mathbf{V}}$ is PSD with eigenvalues $\bar{\lambda}_1 > \bar{\lambda}_2 \geq \dots \geq \bar{\lambda}_n \geq 0$. We use $\bar{\mathbf{x}}$ (a unit vector) to represent the eigenvector of $\bar{\mathbf{V}}$ that corresponds to $\bar{\lambda}_1$.*

In the following, it is useful to think of $\bar{\mathbf{x}}$ is being close to the range of the generative model $G$. In the special case of (3), letting $m$ be the number of samples, it is natural to derive that the upper bound of $\|\mathbf{E}\|_{2\to2}$ grows linearly in $(n/m)^b$ for some positive constant $b$ such as $\frac{1}{2}$ or $1$ (with high probability; see, e.g., (Vershynin, 2010, Corollary 5.35)). In the following, we consider general scenarios with $\mathbf{V}$ depending on $m$ samples (see below for specific examples). Similarly to (Yuan & Zhang, 2013), we may consider a restricted version of $\|\mathbf{E}\|_{2\to2}$, leading to the following.

**Assumption 2** (Assumption on $\mathbf{E}$). *Let $S_1, S_2$ be two (arbitrary) finite sets in $\mathbb{R}^n$ satisfying $m = \Omega(\log(|S_1| \cdot |S_2|))$. Then, we have for all $\mathbf{s}_1 \in S_1$ and $\mathbf{s}_2 \in S_2$ that*

$$\left| \mathbf{s}_1^T \mathbf{E} \mathbf{s}_2 \right| \leq C \sqrt{\frac{\log(|S_1| \cdot |S_2|)}{m}} \cdot \|\mathbf{s}_1\|_2 \cdot \|\mathbf{s}_2\|_2, \tag{7}$$

*where $C$ is an absolute constant. In addition, we have $\|\mathbf{E}\|_{2\to2} = O(n/m)$.*[5]

---

[4]That is, for any $\mathbf{x} \in \mathbb{R}^n$, $\mathcal{P}_G(\mathbf{x}) := \arg\min_{\mathbf{w}\in\text{Range}(G)} \|\mathbf{w} - \mathbf{x}\|_2$. We will implicitly assume that the projection step can be performed accurately, e.g., (Deshpande et al., 2014; Shah & Hegde, 2018; Peng et al., 2020), though in practice approximate methods might be needed, e.g., via gradient descent (Shah & Hegde, 2018) or GAN-based projection methods (Raj et al., 2019).

[5]For the spectral norm of $\mathbf{E}$, one often expects an even tighter bound $O(\sqrt{n/m})$, but we use $O(n/m)$ to simplify the analysis of our examples. Moreover, at least under the typical scaling where $L$ is polynomial

The following examples show that when the number of measurements is sufficiently large, the data matrices corresponding to certain spiked matrix and phase retrieval models satisfy the above assumptions with high probability. Short proofs are given in Appendix B for completeness.

**Example 1** (Spiked covariance model). *In the spiked covariance model (Johnstone & Lu, 2009; Deshpande & Montanari, 2016), the observed vectors $\mathbf{x}_1, \mathbf{x}_2, \ldots, \mathbf{x}_m \in \mathbb{R}^n$ are of the form*

$$\mathbf{x}_i = \sum_{q=1}^r \sqrt{\beta_q} u_{q,i} \mathbf{s}_q + \mathbf{z}_i, \tag{8}$$

*where $\mathbf{s}_1, \ldots, \mathbf{s}_r \in \mathbb{R}^n$ are orthonormal vectors that we want to estimate, while $\mathbf{z}_i \sim \mathcal{N}(\mathbf{0}, \mathbf{I}_n)$ and $u_{q,i} \sim \mathcal{N}(0,1)$ are independent and identically distributed. In addition, $\beta_1, \ldots, \beta_r$ are positive constants that dictate the signal-to-noise ratio (SNR). To simplify the exposition, we focus on the rank-one case and drop the subscript $q \in [r]$. Let*

$$\mathbf{V} = \frac{1}{m} \sum_{i=1}^m (\mathbf{x}_i \mathbf{x}_i^T - \mathbf{I}_n), \tag{9}$$

*and $\bar{\mathbf{V}} = \mathbb{E}[\mathbf{V}] = \beta \mathbf{s}\mathbf{s}^T$.[6] Then, $\bar{\mathbf{V}}$ satisfies Assumption 1 with $\bar{\lambda}_1 = \beta > 0$, $\bar{\lambda}_2 = \ldots = \bar{\lambda}_n = 0$, and $\bar{\mathbf{x}} = \mathbf{s}$. In addition, letting $\mathbf{E} = \mathbf{V} - \bar{\mathbf{V}}$, the Bernstein-type inequality (Vershynin, 2010, Proposition 5.10) for the sum of sub-exponential random variables yields that for any finite sets $S_1, S_2 \subset \mathbb{R}^n$, when $m = \Omega(\log(|S_1| \cdot |S_2|))$, with probability $1 - e^{-\tilde{\Omega}(\log(|S_1| \cdot |S_2|))}$, $\mathbf{E}$ satisfies (7) in Assumption 2. Moreover, standard concentration arguments give $\|\mathbf{E}\|_{2 \to 2} = O(n/m)$ with probability $1 - e^{-\Omega(n)}$.*

**Remark 3.** *We can also consider the simpler spiked Wigner model (Perry et al., 2018; Chung & Lee, 2019) where $\mathbf{V} = \beta \mathbf{s}\mathbf{s}^T + \frac{1}{\sqrt{n}}\mathbf{H}$, with the signal $\mathbf{s}$ being a unit vector, $\beta > 0$ being an SNR parameter, and $\mathbf{H} \in \mathbb{R}^{n \times n}$ being a symmetric matrix with entries drawn i.i.d. (up to symmetry) from $\mathcal{N}(0,1)$. In this case, when $m = n$ is sufficiently large, with high probability, $\bar{\mathbf{V}} := \mathbb{E}[\mathbf{V}] = \beta \mathbf{s}\mathbf{s}^T$ and $\mathbf{E} := \mathbf{V} - \bar{\mathbf{V}}$ similarly satisfy Assumptions 1 and 2 respectively.*

**Example 2** (Phase retrieval). *Let $\mathbf{A} \in \mathbb{R}^{m \times n}$ be a matrix having i.i.d. $\mathcal{N}(0,1)$ entries, and let $\mathbf{a}_i^T$ be the $i$-th row of $\mathbf{A}$. For some unit vector $\mathbf{s}$, suppose that the observed vector is $\mathbf{y} = |\mathbf{A}\mathbf{s}|$, where the absolute value is applied element-wise.[7] We construct the weighted empirical covariance matrix as follows (Zhang et al., 2017; Liu et al., 2021b):*

$$\mathbf{V} = \frac{1}{m} \sum_{i=1}^m \left( y_i \mathbf{a}_i \mathbf{a}_i^T \mathbf{1}_{\{l < y_i < u\}} - \gamma \mathbf{I}_n \right), \tag{10}$$

*where $u > l > 1$ are positive constants, and for $g \sim \mathcal{N}(0,1)$, $\gamma := \mathbb{E}\big[|g|\mathbf{1}_{\{l < |g| < u\}}\big]$. Let $\bar{\mathbf{V}} = \mathbb{E}[\mathbf{V}] = \beta \mathbf{s}\mathbf{s}^T$, where $\beta := \mathbb{E}\big[(|g|^3 - |g|)\mathbf{1}_{\{l < |g| < u\}}\big]$. Then, $\bar{\mathbf{V}}$ satisfies Assumption 1 with $\bar{\lambda}_1 = \beta > 0$, $\bar{\lambda}_2 = \ldots = \bar{\lambda}_n = 0$, and $\bar{\mathbf{x}} = \mathbf{s}$. In addition, letting $\mathbf{E} = \mathbf{V} - \bar{\mathbf{V}}$, we have similarly to Example 1 that $\mathbf{E}$ satisfies Assumption 2 with high probability.*

## 4  MAIN RESULTS

The following theorem concerns globally optimal solutions of (5). The proof is given in Appendix D.

**Theorem 1.** *Let $\mathbf{V} = \bar{\mathbf{V}} + \mathbf{E}$ with Assumptions 1 and 2 being satisfied by $\bar{\mathbf{V}}$ and $\mathbf{E}$ respectively, and let $\mathbf{x}_G := \mathcal{P}_G(\bar{\mathbf{x}}) = \arg\min_{\mathbf{w} \in \text{Range}(G)} \|\mathbf{w} - \bar{\mathbf{x}}\|_2$. Suppose that $\hat{\mathbf{v}}$ is a globally optimal solution to (5). Then, for any $\delta \in (0,1)$, we have*

$$\|\hat{\mathbf{v}}\hat{\mathbf{v}}^T - \bar{\mathbf{x}}\bar{\mathbf{x}}^T\|_{\mathrm{F}} = \frac{O\left(\sqrt{\frac{k \log \frac{Lr}{\delta}}{m}}\right)}{\bar{\lambda}_1 - \bar{\lambda}_2} + O\left(\sqrt{\frac{\delta n/m}{\bar{\lambda}_1 - \bar{\lambda}_2}}\right) + O\left(\sqrt{\frac{(\bar{\lambda}_1 + \epsilon_n)\|\bar{\mathbf{x}} - \mathbf{x}_G\|_2}{\bar{\lambda}_1 - \bar{\lambda}_2}}\right), \tag{11}$$

*where $\epsilon_n = O\big(\sqrt{\frac{k \log \frac{Lr}{\delta}}{m}}\big)$.*

---

in $n$ (Bora et al., 2017), the upper bound for $\|\mathbf{E}\|_{2 \to 2}$ can be easily relaxed to $O\big((n/m)^b\big)$ for any positive constant $b$, without affecting the scaling of our derived statistical rate.

[6]To avoid non-essential complications, $\beta$ is typically assumed to be known (Johnstone & Lu, 2009).

[7]Without loss of generality, we assume that $\mathbf{s}$ is a unit vector. For a general signal $\mathbf{s}$, we may instead focus on estimating $\bar{\mathbf{s}} = \mathbf{s}/\|\mathbf{s}\|_2$, and simply use $\frac{1}{m}\sum_{i=1}^m y_i$ to approximate $\|\mathbf{s}\|_2$.

We have stated this result as an upper bound on $\|\hat{\mathbf{v}}\hat{\mathbf{v}}^T - \bar{\mathbf{x}}\bar{\mathbf{x}}^T\|_{\mathrm{F}}$, which intuitively measures the distance between the 1D subspaces spanned by $\hat{\mathbf{v}}$ and $\bar{\mathbf{x}}$. Note, however, that for any two unit vectors $\mathbf{w}_1, \mathbf{w}_2$ with $\mathbf{w}_1^T\mathbf{w}_2 \geq 0$, the distances $\|\mathbf{w}_1 - \mathbf{w}_2\|_2$ and $\|\mathbf{w}_1\mathbf{w}_1^T - \mathbf{w}_2\mathbf{w}_2^T\|_{\mathrm{F}}$ are equivalent up to constant factors, whereas if $\mathbf{w}_1^T\mathbf{w}_2 \leq 0$, then a similar statement holds for $\|\mathbf{w}_1 + \mathbf{w}_2\|_2$. See Appendix C for the precise statements.

In Theorem 1, the final term quantifies the effect of representation error. When there is no such error (i.e., $\bar{\mathbf{x}} \in \mathrm{Range}(G)$), under the scaling $\bar{\lambda}_1 - \bar{\lambda}_2 = \Theta(1)$, $L = n^{\Omega(1)}$, and $\delta = O(1/n)$, Theorem 1 simplifies to $\|\hat{\mathbf{v}}\hat{\mathbf{v}}^T - \bar{\mathbf{x}}\bar{\mathbf{x}}^T\|_{\mathrm{F}} = O(\sqrt{\frac{k\log L}{m}})$. This provides a natural counterpart to the statistical rate of order $\sqrt{\frac{s\log n}{m}}$ for SPCA mentioned in Section 1.1.

Before providing our main theorem for PPower described in (6), we present the following important lemma, whose proof is presented in Appendix E. To simplify the statement of the lemma, we fix $\delta$ to be $O(1/n)$, though a more general form analogous to Theorem 1 is also possible.

**Lemma 2.** *Let* $\mathbf{V} = \bar{\mathbf{V}} + \mathbf{E}$ *with Assumptions 1 and 2 being satisfied by* $\bar{\mathbf{V}}$ *and* $\mathbf{E}$ *respectively, and further assume that* $\bar{\mathbf{x}} \in \mathrm{Range}(G)$. *Let* $\bar{\gamma} = \bar{\lambda}_2/\bar{\lambda}_1$ *with* $\bar{\lambda}_1 = \Theta(1)$. *Then, for all* $\mathbf{s} \in \mathrm{Range}(G)$ *satisfying* $\mathbf{s}^T\bar{\mathbf{x}} > 0$, *we have*

$$\|\mathcal{P}_G(\mathbf{V}\mathbf{s}) - \bar{\mathbf{x}}\|_2 \leq \frac{2\bar{\gamma}\|\mathbf{s} - \bar{\mathbf{x}}\|_2}{\mathbf{s}^T\bar{\mathbf{x}}} + \frac{C}{\mathbf{s}^T\bar{\mathbf{x}}}\sqrt{\frac{k\log(nLr)}{m}}, \qquad (12)$$

*where $C$ is an absolute constant.*

**Remark 4.** *The assumption* $\mathbf{s}^T\bar{\mathbf{x}} > 0$ *will be particularly satisfied when the range of $G$ only contains nonnegative vectors. As mentioned in various works studying nonnegative SPCA (Zass & Shashua, 2007; Sigg & Buhmann, 2008; Asteris et al., 2014), for several practical fields such as economics, bioinformatics, and computer vision, it is natural to assume that the underlying signal has no negative entries. More generally, the assumption* $\mathbf{s}^T\bar{\mathbf{x}} > 0$ *can be removed if we additionally have that* $-\bar{\mathbf{x}}$ *is also contained in the range of $G$. For this case, when* $\mathbf{s}^T\bar{\mathbf{x}} < 0$, *we can instead derive an upper bound for* $\|\hat{\mathbf{s}} + \bar{\mathbf{x}}\|_2$.

Based on Lemma 2, we have the following theorem, whose proof is given in Appendix F.

**Theorem 2.** *Suppose that the assumptions on the data model* $\mathbf{V} = \bar{\mathbf{V}} + \mathbf{E}$ *are the same as those in Lemma 2, and assume that there exists* $t_0 \in \mathbb{N}$ *such that* $\bar{\mathbf{x}}^T\mathbf{w}^{(t_0)} = 2\bar{\gamma} + \nu$ *with* $2\bar{\gamma} + \nu \leq 1 - \tau$, *where* $\bar{\gamma} = \bar{\lambda}_2/\bar{\lambda}_1 \in [0, 1)$, *and* $\nu, \tau$ *are both positive and scale as* $\Theta(1)$. *Let* $\mu_0 = \frac{2\bar{\gamma}}{\bar{\mathbf{x}}^T\mathbf{w}^{(t_0)}} = \frac{2\bar{\gamma}}{2\bar{\gamma} + \nu} < 1$, *and in addition, suppose that* $m \geq C_{\nu,\tau} \cdot k\log(nLr)$ *with* $C_{\nu,\tau} > 0$ *being large enough. Then, we have after* $\Delta_0 = O\big(\log\big(\frac{m}{k\log(nLr)}\big)\big)$ *iterations of PPower (beyond $t_0$) that*

$$\|\mathbf{w}^{(t)} - \bar{\mathbf{x}}\|_2 \leq \frac{C}{(1 - \mu_0)\nu}\sqrt{\frac{k\log(nLr)}{m}}, \qquad (13)$$

*i.e., this equation holds for all* $t \geq T_0 := t_0 + \Delta_0$. *Moreover, if* $\bar{\gamma} = 0$ *then* $\Delta_0 \leq 1$, *whereas if* $\bar{\gamma} = \Theta(1)$, *we have exponentially fast convergence via the following contraction property: There exists a constant* $\xi \in (0, 1)$ *such that for* $t \in [t_0, T_0)$, *it holds that*

$$\|\mathbf{w}^{(t+1)} - \bar{\mathbf{x}}\|_2 \leq (1 - \xi)\|\mathbf{w}^{(t)} - \bar{\mathbf{x}}\|_2. \qquad (14)$$

Regarding the assumption $\bar{\mathbf{x}}^T\mathbf{w}^{(t_0)} \geq 2\bar{\gamma} + \nu$, we note that when $t_0 = 0$, this condition can be viewed as having a good initialization. For both Examples 1 and 2, we have $\bar{\gamma} = 0$. Thus, for the spiked covariance and phase retrieval models corresponding to these examples, the assumption $\bar{\mathbf{x}}^T\mathbf{w}^{(t_0)} \geq 2\bar{\gamma} + \nu$ reduces to $\bar{\mathbf{x}}^T\mathbf{w}^{(t_0)} \geq \nu$ for a sufficiently small positive constant $\nu$, which results in a mild assumption. Such an assumption is also required for the projected power method devised for cone-constrained PCA under the simple spiked Wigner model, with the underlying signal being assumed to lie in a convex cone; see (Deshpande et al., 2014, Theorem 3). Despite using a similar assumption on the initialization, our proof techniques are significantly different from Deshpande et al. (2014); see Appendix G for discussion.

When $L$ is polynomial in $n$, Theorem 2 reveals that we have established conditions under which PPower in (6) converges exponentially fast to a point achieving the statistical rate of order $\sqrt{\frac{k\log L}{m}}$.

Based on the minimax rates for SPCA (Vu & Lei, 2012; Birnbaum et al., 2013) and the information-theoretic lower bounds for CS with generative models (Liu & Scarlett, 2020b; Kamath et al., 2020), the optimal rate for GPCA is naturally conjectured to be of the same order $\sqrt{\frac{k \log L}{m}}$. We highlight that Theorem 2 partially addresses the computational-to-statistical gap (e.g., see (Wang et al., 2016; Hand et al., 2018; Aubin et al., 2019; Cocola et al., 2020)) for spiked matrix recovery and phase retrieval under a generative prior, though closing it completely would require efficiently finding a good initialization and addressing the assumption of exact projections.

Perhaps the main caveat to Theorem 2 is that it assumes the projection step can be performed exactly. However, this is a standard assumption in analyses of projected gradient methods, e.g., see (Shah & Hegde, 2018), and both gradient-based projection and GAN-based projection have been shown to be highly effective in practice (Shah & Hegde, 2018; Raj et al., 2019).

## 5 EXPERIMENTS

In this section, we experimentally study the performance of Algorithm 1 (`PPower`). We note that these experiments are intended as a simple proof of concept rather than seeking to be comprehensive, as our contributions are primarily theoretical. We compare with the truncated power method (`TPower`) devised for SPCA proposed in (Yuan & Zhang, 2013, Algorithm 1) and the vanilla power method (`Power`) that performs the iterative procedure $\mathbf{w}^{(t+1)} = (\mathbf{V}\mathbf{w}^{(t)})/\|\mathbf{V}\mathbf{w}^{(t)}\|_2$. For a fair comparison, for `PPower`, `TPower`, and `Power`, we use the same initial vector. Specifically, as mentioned in (Liu et al., 2021b, Section V), we choose the initialization vector $\mathbf{w}^{(0)}$ as the column of $\mathbf{V}$ that corresponds to its largest diagonal entry. For all three algorithms, the total number of iterations $T$ is set to be 30. To compare the performance across algorithms, we use the scale-invariant Cosine Similarity metric defined as $\text{Cossim}\left(\bar{\mathbf{x}}, \mathbf{w}^{(T)}\right) := \frac{\langle \bar{\mathbf{x}}, \mathbf{w}^{(T)} \rangle}{\|\bar{\mathbf{x}}\|_2 \|\mathbf{w}^{(T)}\|_2}$, where $\bar{\mathbf{x}}$ is the ground-truth signal to estimate, and $\mathbf{w}^{(T)}$ denotes the output vector of the algorithm.

The experiments are performed on the MNIST (LeCun et al., 1998), Fashion-MNIST (Xiao et al., 2017) and CelebA (Liu et al., 2015) datasets, with the numerical results for the Fashion-MNIST and CelebA datasets being presented in Appendix H and I. The MNIST dataset consists of $60,000$ images of handwritten digits. The size of each image is $28 \times 28$, and thus $n = 784$. To reduce the impact of local minima, we perform 10 random restarts, and choose the best among these. The cosine similarity is averaged over the test images, and also over these 10 random restarts. The generative model $G$ is set to be a pre-trained variational autoencoder (VAE) model with latent dimension $k = 20$. We use the VAE model trained by the authors of (Bora et al., 2017) directly, for which the encoder and decoder are both fully connected neural networks with two hidden layers, with the architecture being $20 - 500 - 500 - 784$. The VAE is trained by the Adam optimizer with a mini-batch size of 100 and a learning rate of 0.001. The projection step $\mathcal{P}_G(\cdot)$ is solved by the Adam optimizer with a learning rate of 0.03 and 200 steps. In each iteration of `TPower`, the calculated entries are truncated to zero except for the largest $q$ entries, where $q \in \mathbb{N}$ is a tuning parameter. Since for `TPower`, $q$ is usually selected as an integer larger than the true sparsity level, and since it is unlikely that the image of the MNIST dataset can be well approximated by a $k$-sparse vector with $k = 20$, we choose a relatively large $q$, namely $q = 150$. Similarly to (Bora et al., 2017) and other related works, we only report the results on a test set that is unseen by the pre-trained VAE model, i.e., the training of $G$ and the PPower computations do not use common data.[8]

1. **Spiked covariance model (Example 1)**: The numerical results are shown in Figures 1 and 2. We observe from Figure 1 that `Power` and `TPower` attain poor reconstructions, and the generative prior based method `PPower` attains significantly better reconstructions. To illustrate the effect of the sample size $m$, we fix the SNR parameter $\beta = 1$ and vary $m$ in $\{100, 200, 300, 400, 500\}$. In addition, to illustrate the effect of the SNR parameter $\beta$, we fix $m = 300$, and vary $\beta$ in $\{0.6, 0.7, 0.8, 0.9, 1, 2, 3, 4\}$. From Figure 2, we observe that for these settings of $m$ and $\beta$, `PPower` always leads to a much higher cosine similarity compared to `Power` and `TPower`, which is natural given the more precise modeling assumptions used.

---

[8]All experiments are run using Python 3.6 and Tensorflow 1.5.0, with a NVIDIA GeForce GTX 1080 Ti 11GB GPU. The corresponding code is available at `https://github.com/liuzq09/GenerativePCA`.

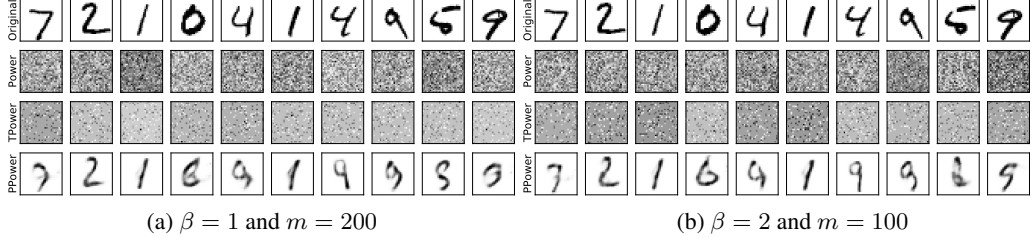

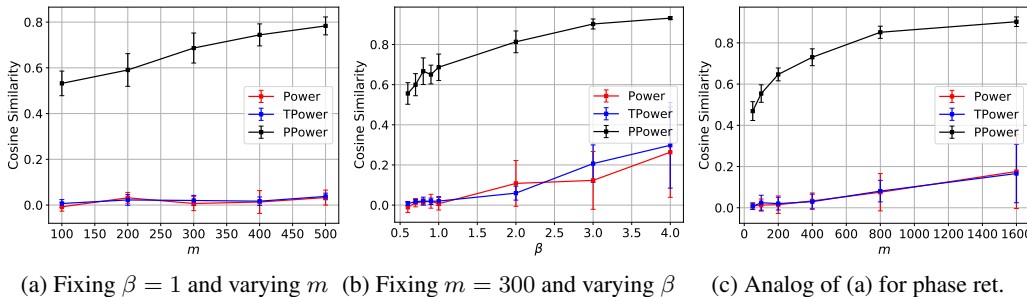

(a) $\beta = 1$ and $m = 200$            (b) $\beta = 2$ and $m = 100$

Figure 1: Examples of reconstructed images of the MNIST dataset for the spiked covariance model.

(a) Fixing $\beta = 1$ and varying $m$    (b) Fixing $m = 300$ and varying $\beta$    (c) Analog of (a) for phase ret.

Figure 2: Quantitative comparisons of the performance of `Power`, `TPower` and `PPower` according to the Cosine Similarity for the MNIST dataset, under both the spiked covariance model (Left/Middle) and phase retrieval model (Right).

2. **Phase retrieval (Example 2)**: The results are shown in Figure 2 (Right) and Figure 3. Again, we can observe that `PPower` significantly outperforms `Power` and `TPower`. In particular, for sparse phase retrieval, when performing experiments on image datasets, even for the noiseless setting, solving an eigenvalue problem similar to (5) can typically only serve as a spectral initialization step, with a subsequent iterative algorithm being required to refine the initial guess. In view of this, it is notable that for phase retrieval with generative priors, `PPower` can return meaningful reconstructed images for $m = 200$, which is small compared to $n = 784$.

## 6 CONCLUSION

We have proposed a quadratic estimator for eigenvalue problems with generative models, and we showed that this estimator attains a statistical rate of order $\sqrt{\frac{k \log L}{m}}$. We provided a projected power method to efficiently solve (modulo the complexity of the projection step) the corresponding optimization problem, and showed that our method converges exponentially fast to a point achieving a statistical rate of order $\sqrt{\frac{k \log L}{m}}$ under suitable conditions.

**Acknowledgment.** J.S. was supported by the Singapore National Research Foundation (NRF) under grant R-252-000-A74-281, and S.G. was supported in part by the MOE grants R-146-000-250-133, R-146-000-312-114, and MOE-T2EP20121-0013.

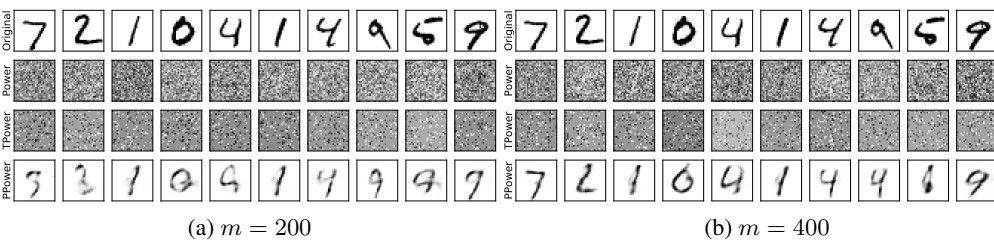

(a) $m = 200$            (b) $m = 400$

Figure 3: Examples of reconstructed images of the MNIST dataset for phase retrieval.

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

## APPENDIX (GENERATIVE PRINCIPAL COMPONENT ANALYSIS, LIU/LIU/GHOSH/HAN/SCARLETT, ICLR 2022)

## A    PROOF OF LEMMA 1 (NON-DECREASING PROPERTY OF $Q$)

Since $\mathbf{w}^{(t+1)} = \mathcal{P}_G\left(\mathbf{V}\mathbf{w}^{(t)}\right)$ and $\mathbf{w}^{(t)} \in \mathrm{Range}(G)$, we have

$$\|\mathbf{V}\mathbf{w}^{(t)} - \mathbf{w}^{(t+1)}\|_2 \le \|\mathbf{V}\mathbf{w}^{(t)} - \mathbf{w}^{(t)}\|_2, \tag{15}$$

and since $\|\mathbf{w}^{(t+1)}\|_2 = \|\mathbf{w}^{(t)}\|_2 = 1$, expanding the square gives

$$\left\langle \mathbf{V}\mathbf{w}^{(t)}, \mathbf{w}^{(t+1)} \right\rangle \ge Q(\mathbf{w}^{(t)}). \tag{16}$$

Then, we obtain

$$Q(\mathbf{w}^{(t+1)}) = \left\langle \mathbf{V}\mathbf{w}^{(t+1)}, \mathbf{w}^{(t+1)} \right\rangle \tag{17}$$

$$= Q(\mathbf{w}^{(t+1)} - \mathbf{w}^{(t)}) + 2\left\langle \mathbf{V}(\mathbf{w}^{(t+1)} - \mathbf{w}^{(t)}), \mathbf{w}^{(t)} \right\rangle + Q(\mathbf{w}^{(t)}) \tag{18}$$

$$\ge 2\left\langle \mathbf{V}(\mathbf{w}^{(t+1)} - \mathbf{w}^{(t)}), \mathbf{w}^{(t)} \right\rangle + Q(\mathbf{w}^{(t)}) \tag{19}$$

$$\ge Q(\mathbf{w}^{(t)}), \tag{20}$$

where (18) follows by writing $\mathbf{w}^{(t+1)} = \mathbf{w}^{(t)} + (\mathbf{w}^{(t+1)} - \mathbf{w}^{(t)})$ and expanding, (19) follows from the assumption that $\mathbf{V}$ is PSD, and (20) follows from (16).

## B    PROOFS FOR SPIKED MATRIX AND PHASE RETRIEVAL EXAMPLES

Before proceeding, we present the following standard definitions.

**Definition 1.** *A random variable $X$ is said to be sub-Gaussian if there exists a positive constant $C$ such that $(\mathbb{E}\left[|X|^p\right])^{1/p} \le C\sqrt{p}$ for all $p \ge 1$. The sub-Gaussian norm of a sub-Gaussian random variable $X$ is defined as $\|X\|_{\psi_2} := \sup_{p \ge 1} p^{-1/2}\left(\mathbb{E}\left[|X|^p\right]\right)^{1/p}$.*

**Definition 2.** *A random variable $X$ is said to be sub-exponential if there exists a positive constant $C$ such that $(\mathbb{E}\left[|X|^p\right])^{\frac{1}{p}} \le Cp$ for all $p \ge 1$. The sub-exponential norm of $X$ is defined as $\|X\|_{\psi_1} := \sup_{p \ge 1} p^{-1}\left(\mathbb{E}\left[|X|^p\right]\right)^{\frac{1}{p}}$.*

The following lemma states that the product of two sub-Gaussian random variables is sub-exponential, regardless of the dependence between them.

**Lemma 3.** *(Vershynin, 2018, Lemma 2.7.7) Let $X$ and $Y$ be sub-Gaussian random variables (not necessarily independent). Then $XY$ is sub-exponential, and satisfies*

$$\|XY\|_{\psi_1} \le \|X\|_{\psi_2}\|Y\|_{\psi_2}. \tag{21}$$

The following lemma provides a useful concentration inequality for the sum of independent sub-exponential random variables.

**Lemma 4.** *(Vershynin, 2010, Proposition 5.16) Let $X_1, \ldots, X_N$ be independent zero-mean sub-exponential random variables, and $K = \max_i \|X_i\|_{\psi_1}$. Then for every $\boldsymbol{\alpha} = [\alpha_1, \ldots, \alpha_N]^T \in \mathbb{R}^N$ and $\epsilon \ge 0$, it holds that*

$$\mathbb{P}\left(\Big|\sum_{i=1}^N \alpha_i X_i\Big| \ge \epsilon\right) \le 2\exp\left(-c \cdot \min\Big(\frac{\epsilon^2}{K^2\|\boldsymbol{\alpha}\|_2^2}, \frac{\epsilon}{K\|\boldsymbol{\alpha}\|_\infty}\Big)\right), \tag{22}$$

*where $c > 0$ is an absolute constant. In particular, with $\boldsymbol{\alpha} = \left[\frac{1}{N}, \ldots, \frac{1}{N}\right]^T$, we have*

$$\mathbb{P}\left(\Big|\frac{1}{N}\sum_{i=1}^N X_i\Big| \ge \epsilon\right) \le 2\exp\left(-c \cdot \min\Big(\frac{N\epsilon^2}{K^2}, \frac{N\epsilon}{K}\Big)\right). \tag{23}$$

The widely-used notion of an $\epsilon$-net is introduced as follows.

**Definition 3.** *Let $(\mathcal{X}, d)$ be a metric space, and fix $\epsilon > 0$. A subset $S \subseteq \mathcal{X}$ is said be an $\epsilon$-net of $\mathcal{X}$ if, for all $x \in \mathcal{X}$, there exists some $s \in S$ such that $d(s, x) \le \epsilon$. The minimal cardinality of an $\epsilon$-net of $\mathcal{X}$, if finite, is denoted $\mathcal{C}(\mathcal{X}, \epsilon)$ and is called the covering number of $\mathcal{X}$ (at scale $\epsilon$).*

The following lemma provides a useful upper bound for the covering number of the unit sphere.

**Lemma 5.** (Vershynin, 2010, Lemma 5.2) *The unit Euclidean sphere $\mathcal{S}^{N-1}$ equipped with the Euclidean metric satisfies for every $\epsilon > 0$ that*

$$\mathcal{C}(\mathcal{S}^{N-1}, \epsilon) \le \left(1 + \frac{2}{\epsilon}\right)^N. \tag{24}$$

The following lemma provides an upper bound for the spectral norm of a symmetric matrix.

**Lemma 6.** (Vershynin, 2010, Lemma 5.4) *Let $\mathbf{X}$ be a symmetric $N \times N$ matrix, and let $\mathcal{C}_\epsilon$ be an $\epsilon$-net of $\mathcal{S}^{N-1}$ for some $\epsilon \in [0, 1/2]$. Then,*

$$\|\mathbf{X}\|_{2 \to 2} = \sup_{\mathbf{r} \in \mathcal{S}^{N-1}} |\langle \mathbf{X}\mathbf{r}, \mathbf{r} \rangle| \le (1 - 2\epsilon)^{-1} \sup_{\mathbf{r} \in \mathcal{C}_\epsilon} |\langle \mathbf{X}\mathbf{r}, \mathbf{r} \rangle|. \tag{25}$$

With the above auxiliary results in place, we provide the proofs of Assumption 2 holding for the two examples described in Section 3.

### B.1 SPIKED COVARIANCE MODEL (EXAMPLE 1)

As per Assumption 2, fix two finite signal sets $S_1$ and $S_2$. For $r = 1$, we have $\mathbf{x}_i = \sqrt{\beta} u_i \mathbf{s} + \mathbf{z}_i$ and a direct calculation gives $\mathbb{E}[\mathbf{V}] = \bar{\mathbf{V}} = \beta \mathbf{s}\mathbf{s}^T$. Recall also that $\|\mathbf{s}\|_2 = 1$, $u_i \sim \mathcal{N}(0, 1)$, and $\mathbf{z}_i \sim \mathcal{N}(\mathbf{0}, \mathbf{I}_n)$. It follows that for any $\mathbf{s}_1 \in S_1$, we have that $\mathbf{s}_1^T \mathbf{x}_i = \sqrt{\beta} u_i \mathbf{s}^T \mathbf{s}_1 + \mathbf{z}_i^T \mathbf{s}_1$ is sub-Gaussian, with the sub-Gaussian norm being upper bounded by $C(\sqrt{\beta} + 1)\|\mathbf{s}_1\|_2$. Similarly, we have for any $\mathbf{s}_2 \in S_2$ that $\|\mathbf{s}_2^T \mathbf{x}_i\|_{\psi_2} \le C(\sqrt{\beta} + 1)\|\mathbf{s}_2\|_2$. Applying Lemma 3, we deduce that $(\mathbf{s}_1^T \mathbf{x}_i)(\mathbf{s}_2^T \mathbf{x}_i)$ is sub-exponential, with the sub-exponential norm being upper bounded by $C^2(\sqrt{\beta} + 1)^2 \|\mathbf{s}_1\|_2 \|\mathbf{s}_2\|_2$. In addition, from (9) and $\bar{\mathbf{V}} = \beta \mathbf{s}\mathbf{s}^T$, we have

$$\mathbf{s}_1^T \mathbf{E}\mathbf{s}_2 = \mathbf{s}_1^T (\mathbf{V} - \bar{\mathbf{V}})\mathbf{s}_2 \tag{26}$$

$$= \frac{1}{m} \sum_{i=1}^{m} \left( (\mathbf{x}_i^T \mathbf{s}_1)(\mathbf{x}_i^T \mathbf{s}_2) - \left((\mathbf{s}_1^T \mathbf{s}_2) + \beta(\mathbf{s}^T \mathbf{s}_1)(\mathbf{s}^T \mathbf{s}_2)\right)\right), \tag{27}$$

and we observe that $\mathbb{E}[(\mathbf{x}_i^T \mathbf{s}_1)(\mathbf{x}_i^T \mathbf{s}_2)] = (\mathbf{s}_1^T \mathbf{s}_2) + \beta(\mathbf{s}^T \mathbf{s}_1)(\mathbf{s}^T \mathbf{s}_2)$. Then, from Lemma 4, we obtain that for any $t > 0$ satisfying $m = \Omega(t)$, the following holds with probability $1 - e^{-\Omega(t)}$ (recall that $C$ may vary from line to line):

$$\left| \frac{1}{m} \sum_{i=1}^{m} \left( (\mathbf{x}_i^T \mathbf{s}_1)(\mathbf{x}_i^T \mathbf{s}_2) - \left((\mathbf{s}_1^T \mathbf{s}_2) + \beta(\mathbf{s}^T \mathbf{s}_1)(\mathbf{s}^T \mathbf{s}_2)\right)\right)\right| \le C(\sqrt{\beta} + 1)^2 \|\mathbf{s}_1\|_2 \|\mathbf{s}_2\|_2 \cdot \frac{\sqrt{t}}{\sqrt{m}}, \tag{28}$$

where we note that the assumption $m = \Omega(t)$ ensures that the first term is dominant in the minimum in (23). Taking a union bound over all $\mathbf{s}_1 \in S_1$ and $\mathbf{s}_2 \in S_2$, and setting $t = \log(|S_1| \cdot |S_2|)$, we obtain with probability $1 - e^{-\Omega(\log(|S_1| \cdot |S_2|))}$ that (7) holds (with $\beta$ being a fixed positive constant).

Next, we bound $|\mathbf{r}^T \mathbf{E}\mathbf{r}|$ for fixed $\mathbf{r} \in \mathcal{S}^{n-1}$, but this time consider $t > 0$ (different from the above $t$) satisfying $t = \Omega(m)$. In this case, we can follow the above analysis (with $\mathbf{s}_1$ and $\mathbf{s}_2$ both replaced by $\mathbf{r}$), but the assumption $t = \Omega(m)$ means that when applying Lemma 4, the second term in the minimum in (23) is now the dominant one. As a result, for any $t > 0$ satisfying $t = \Omega(m)$, and any $\mathbf{r} \in \mathcal{S}^{n-1}$, we have with probability $1 - e^{-\Omega(t)}$ that

$$|\mathbf{r}^T \mathbf{E}\mathbf{r}| = \left| \frac{1}{m} \sum_{i=1}^{m} \left( (\mathbf{x}_i^T \mathbf{r})^2 - \left(1 + \beta(\mathbf{s}^T \mathbf{r})^2\right)\right)\right| \tag{29}$$

$$\le C(\sqrt{\beta} + 1) \cdot \frac{t}{m}. \tag{30}$$

From Lemma 5, there exists an $(1/4)$-net $\mathcal{C}_{\frac{1}{4}}$ of $\mathcal{S}^{n-1}$ satisfying $\log \left|\mathcal{C}_{\frac{1}{4}}\right| \leq n \log 9$. Taking a union bound over all $\mathbf{r} \in \mathcal{C}_{\frac{1}{4}}$, and setting $t = Cn$, we obtain with probability $1 - e^{-\Omega(n)}$ that

$$\sup_{\mathbf{r} \in \mathcal{C}_{\frac{1}{4}}} \left|\mathbf{r}^T \mathbf{E} \mathbf{r}\right| = O\left(\frac{n}{m}\right). \tag{31}$$

Then, from Lemma 6, we have

$$\|\mathbf{E}\|_{2 \to 2} \leq 2 \sup_{\mathbf{r} \in \mathcal{C}_{\frac{1}{4}}} \left|\mathbf{r}^T \mathbf{E} \mathbf{r}\right| = O\left(\frac{n}{m}\right). \tag{32}$$

## B.2 Phase Retrieval (Example 2)

Let $\mathbf{W} = \frac{1}{m} \sum_{i=1}^{m} y_i \mathbf{a}_i \mathbf{a}_i^T \mathbf{1}_{\{l < y_i < u\}}$ and $\bar{\mathbf{W}} = \beta \mathbf{s} \mathbf{s}^T + \gamma \mathbf{I}_n$. It is shown in (Liu et al., 2021b, Lemma 8) that

$$\mathbb{E}[\mathbf{W}] = \bar{\mathbf{W}}, \tag{33}$$

which implies

$$\mathbb{E}[\mathbf{V}] = \beta \mathbf{s} \mathbf{s}^T = \bar{\mathbf{V}}. \tag{34}$$

Then, for any $\mathbf{s}_1 \in S_1$ and $\mathbf{s}_2 \in S_2$, we have

$$\mathbf{s}_1^T \mathbf{E} \mathbf{s}_2 = \mathbf{s}_1^T (\mathbf{V} - \bar{\mathbf{V}}) \mathbf{s}_2 = \mathbf{s}_1^T (\mathbf{W} - \bar{\mathbf{W}}) \mathbf{s}_2 \tag{35}$$

$$= \frac{1}{m} \sum_{i=1}^{m} \left( y_i (\mathbf{a}_i^T \mathbf{s}_1)(\mathbf{a}_i^T \mathbf{s}_2) \mathbf{1}_{\{l < y_i < u\}} - \left( \beta(\mathbf{s}^T \mathbf{s}_1)(\mathbf{s}^T \mathbf{s}_2) + \gamma(\mathbf{s}_1^T \mathbf{s}_2) \right) \right). \tag{36}$$

Since each $\mathbf{a}_i$ has i.i.d. $\mathcal{N}(0,1)$ entries, we observe that $y_i(\mathbf{a}_i^T \mathbf{s}_1)(\mathbf{a}_i^T \mathbf{s}_2) \mathbf{1}_{\{l < y_i < u\}}$ is sub-exponential with the sub-exponential norm being upper bounded by $Cu\|\mathbf{s}_1\|_2\|\mathbf{s}_2\|_2$. In addition, from (33), we have $\mathbb{E}[y_i(\mathbf{a}_i^T \mathbf{s}_1)(\mathbf{a}_i^T \mathbf{s}_2) \mathbf{1}_{\{l < y_i < u\}}] = \beta(\mathbf{s}^T \mathbf{s}_1)(\mathbf{s}^T \mathbf{s}_2) + \gamma(\mathbf{s}_1^T \mathbf{s}_2)$. Then, from Lemma 4, we obtain that for any $t > 0$ satisfying $m = \Omega(t)$, with probability $1 - e^{-\Omega(t)}$,

$$\left| \frac{1}{m} \sum_{i=1}^{m} \left( y_i(\mathbf{a}_i^T \mathbf{s}_1)(\mathbf{a}_i^T \mathbf{s}_2) \mathbf{1}_{\{l < y_i < u\}} - (\beta(\mathbf{s}^T \mathbf{s}_1)(\mathbf{s}^T \mathbf{s}_2) + \gamma(\mathbf{s}_1^T \mathbf{s}_2)) \right) \right| \leq Cu\|\mathbf{s}_1\|_2\|\mathbf{s}_2\|_2 \cdot \frac{\sqrt{t}}{\sqrt{m}}. \tag{37}$$

Taking a union bound over all $\mathbf{s}_1 \in S_1$ and $\mathbf{s}_2 \in S_2$, and setting $t = \log(|S_1| \cdot |S_2|)$, we obtain that with probability $1 - e^{-\Omega(\log(|S_1| \cdot |S_2|))}$, (7) holds as desired (with $u$ being a fixed positive constant). In addition, similarly to (32), we have with probability $1 - e^{-\Omega(n)}$ that $\|\mathbf{E}\|_{2 \to 2} = O\left(\frac{n}{m}\right)$.

## C Equivalence of Distances

The following lemma gives a useful equivalence between two distances.

**Lemma 7.** *For any pair of unit vectors $\mathbf{w}_1, \mathbf{w}_2$ with $\mathbf{w}_1^T \mathbf{w}_2 \geq 0$, we have*

$$\|\mathbf{w}_1 - \mathbf{w}_2\|_2^2 \leq \|\mathbf{w}_1 \mathbf{w}_1^T - \mathbf{w}_2 \mathbf{w}_2^T\|_F^2 \leq 2\|\mathbf{w}_1 - \mathbf{w}_2\|_2^2. \tag{38}$$

*Moreover, if $\mathbf{w}_1^T \mathbf{w}_2 < 0$, then the same holds with $\|\mathbf{w}_1 - \mathbf{w}_2\|_2$ replaced by $\|\mathbf{w}_1 + \mathbf{w}_2\|_2$.*

*Proof.* When $\mathbf{w}_1^T \mathbf{w}_2 \geq 0$, we have

$$\|\mathbf{w}_1 \mathbf{w}_1^T - \mathbf{w}_2 \mathbf{w}_2^T\|_F^2 = \mathrm{tr}((\mathbf{w}_1 \mathbf{w}_1^T - \mathbf{w}_2 \mathbf{w}_2^T)^T (\mathbf{w}_1 \mathbf{w}_1^T - \mathbf{w}_2 \mathbf{w}_2^T)) \tag{39}$$

$$= 2\left(1 - (\mathbf{w}_1^T \mathbf{w}_2)^2\right) \tag{40}$$

$$\geq 2\left(1 - \mathbf{w}_1^T \mathbf{w}_2\right) \tag{41}$$

$$= \|\mathbf{w}_1 - \mathbf{w}_2\|_2^2, \tag{42}$$

where (40) follows by expanding the product and writing $\mathrm{tr}(\mathbf{w}_1 \mathbf{w}_1^T \mathbf{w}_1 \mathbf{w}_1^T) = \mathrm{tr}(\mathbf{w}_1^T \mathbf{w}_1 \mathbf{w}_1^T \mathbf{w}_1) = (\mathbf{w}_1^T \mathbf{w}_1)^2 = 1$ and handling the other terms similarly, and (41) follows since $\mathbf{w}_1^T \mathbf{w}_2 \in (0, 1)$. In

addition, we have

$$\|\mathbf{w}_1\mathbf{w}_1^T - \mathbf{w}_2\mathbf{w}_2^T\|_{\mathrm{F}}^2 = 2\left(1 - (\mathbf{w}_1^T\mathbf{w}_2)^2\right) \tag{43}$$

$$= 2\left(1 - \mathbf{w}_1^T\mathbf{w}_2\right)\left(1 + \mathbf{w}_1^T\mathbf{w}_2\right) \tag{44}$$

$$\leq 4\left(1 - \mathbf{w}_1^T\mathbf{w}_2\right) \tag{45}$$

$$= 2\|\mathbf{w}_1 - \mathbf{w}_2\|_2^2, \tag{46}$$

which gives the desired inequality. The case $\mathbf{w}_1^T\mathbf{w}_2 < 0$ is handled similarly $\qquad\square$

## D  PROOF OF THEOREM 1 (GUARANTEE ON THE GLOBAL OPTIMUM)

Let the singular value decomposition (SVD) of $\bar{\mathbf{V}}$ be

$$\bar{\mathbf{V}} = \bar{\mathbf{U}}\bar{\mathbf{D}}\bar{\mathbf{U}}^T, \tag{47}$$

where $\bar{\mathbf{D}} = \mathrm{Diag}([\bar{\lambda}_1, \ldots, \bar{\lambda}_n])$, and $\bar{\mathbf{U}} \in \mathbb{R}^n$ is an orthonormal matrix with the first column being $\bar{\mathbf{x}}$. For $i > 1$, let the $i$-th column of $\bar{\mathbf{U}}$ be $\bar{\mathbf{u}}_i$. Then, we have

$$\hat{\mathbf{v}}^T\bar{\mathbf{V}}\hat{\mathbf{v}} = \bar{\lambda}_1\left(\bar{\mathbf{x}}^T\hat{\mathbf{v}}\right)^2 + \sum_{i>1}\bar{\lambda}_i\left(\bar{\mathbf{u}}_i^T\hat{\mathbf{v}}\right)^2 \tag{48}$$

$$\leq \bar{\lambda}_1\left(\bar{\mathbf{x}}^T\hat{\mathbf{v}}\right)^2 + \bar{\lambda}_2\sum_{i>1}\left(\bar{\mathbf{u}}_i^T\hat{\mathbf{v}}\right)^2 \tag{49}$$

$$= \bar{\lambda}_1\left(\bar{\mathbf{x}}^T\hat{\mathbf{v}}\right)^2 + \bar{\lambda}_2\left(1 - \left(\bar{\mathbf{x}}^T\hat{\mathbf{v}}\right)^2\right), \tag{50}$$

where we use $\left(\bar{\mathbf{x}}^T\hat{\mathbf{v}}\right)^2 + \sum_{i>1}\left(\bar{\mathbf{u}}_i^T\hat{\mathbf{v}}\right)^2 = 1$ in (50).

In addition, for any $\mathbf{A} \in \mathbb{R}^{n\times n}$ and any $\mathbf{s}_1, \mathbf{s}_2 \in \mathbb{R}^n$, we have

$$\mathbf{s}_1^T\mathbf{A}\mathbf{s}_1 - \mathbf{s}_2^T\mathbf{A}\mathbf{s}_2 = \left(\frac{\mathbf{s}_1 + \mathbf{s}_2}{2} + \frac{\mathbf{s}_1 - \mathbf{s}_2}{2}\right)^T\mathbf{A}\left(\frac{\mathbf{s}_1 + \mathbf{s}_2}{2} + \frac{\mathbf{s}_1 - \mathbf{s}_2}{2}\right)$$
$$- \left(\frac{\mathbf{s}_1 + \mathbf{s}_2}{2} - \frac{\mathbf{s}_1 - \mathbf{s}_2}{2}\right)^T\mathbf{A}\left(\frac{\mathbf{s}_1 + \mathbf{s}_2}{2} - \frac{\mathbf{s}_1 - \mathbf{s}_2}{2}\right) \tag{51}$$

$$= 2\left(\frac{\mathbf{s}_1 + \mathbf{s}_2}{2}\right)^T\mathbf{A}\left(\frac{\mathbf{s}_1 - \mathbf{s}_2}{2}\right) + 2\left(\frac{\mathbf{s}_1 - \mathbf{s}_2}{2}\right)^T\mathbf{A}\left(\frac{\mathbf{s}_1 + \mathbf{s}_2}{2}\right). \tag{52}$$

In particular, when $\mathbf{A}$ is symmetric, we obtain

$$\mathbf{s}_1^T\mathbf{A}\mathbf{s}_1 - \mathbf{s}_2^T\mathbf{A}\mathbf{s}_2 = (\mathbf{s}_1 + \mathbf{s}_2)^T\mathbf{A}(\mathbf{s}_1 - \mathbf{s}_2). \tag{53}$$

Let $M$ be a $(\delta/L)$-net of $B_2^k(r)$; from (Vershynin, 2010, Lemma 5.2), we know that there exists such a net with

$$\log|M| \leq k\log\frac{4Lr}{\delta}. \tag{54}$$

Since $G$ is $L$-Lipschitz continuous, we have that $G(M)$ is a $\delta$-net of $\mathrm{Range}(G) = G(B_2^k(r))$. We write

$$\hat{\mathbf{v}} = (\hat{\mathbf{v}} - \tilde{\mathbf{x}}) + \tilde{\mathbf{x}}, \tag{55}$$

where $\tilde{\mathbf{x}} \in G(M)$ satisfies $\|\hat{\mathbf{v}} - \tilde{\mathbf{x}}\|_2 \leq \delta$. Suppose that $\bar{\mathbf{x}}^T\hat{\mathbf{v}} \geq 0$; if this is not the case, we can use analogous steps to obtain an upper bound for $\|\bar{\mathbf{x}} + \hat{\mathbf{v}}\|_2$ instead of $\|\bar{\mathbf{x}} - \hat{\mathbf{v}}\|_2$. We have

$$\frac{\bar{\lambda}_1 - \bar{\lambda}_2}{2} \cdot \|\bar{\mathbf{x}} - \hat{\mathbf{v}}\|_2^2 \tag{56}$$

$$= \left(\bar{\lambda}_1 - \bar{\lambda}_2\right)\left(1 - \bar{\mathbf{x}}^T\hat{\mathbf{v}}\right) \tag{57}$$

$$\leq \left(\bar{\lambda}_1 - \bar{\lambda}_2\right)\left(1 - \left(\bar{\mathbf{x}}^T\hat{\mathbf{v}}\right)^2\right) \tag{58}$$

$$= \bar{\lambda}_1 - \left(\bar{\lambda}_1\left(\bar{\mathbf{x}}^T\hat{\mathbf{v}}\right)^2 + \bar{\lambda}_2\left(1 - \left(\bar{\mathbf{x}}^T\hat{\mathbf{v}}\right)^2\right)\right) \tag{59}$$

$$= \bar{\mathbf{x}}^T \bar{\mathbf{V}} \bar{\mathbf{x}} - \left( \bar{\lambda}_1 \left( \bar{\mathbf{x}}^T \hat{\mathbf{v}} \right)^2 + \bar{\lambda}_2 \left( 1 - \left( \bar{\mathbf{x}}^T \hat{\mathbf{v}} \right)^2 \right) \right) \tag{60}$$

$$\leq \bar{\mathbf{x}}^T \bar{\mathbf{V}} \bar{\mathbf{x}} - \hat{\mathbf{v}}^T \bar{\mathbf{V}} \hat{\mathbf{v}} \tag{61}$$

$$= \mathbf{x}_G^T \bar{\mathbf{V}} \mathbf{x}_G + (\bar{\mathbf{x}} + \mathbf{x}_G)^T \bar{\mathbf{V}} (\bar{\mathbf{x}} - \mathbf{x}_G) - \hat{\mathbf{v}}^T \bar{\mathbf{V}} \hat{\mathbf{v}} \tag{62}$$

$$\leq \mathbf{x}_G^T \bar{\mathbf{V}} \mathbf{x}_G + 2\bar{\lambda}_1 \|\bar{\mathbf{x}} - \mathbf{x}_G\|_2 - \hat{\mathbf{v}}^T \bar{\mathbf{V}} \hat{\mathbf{v}} \tag{63}$$

$$= \mathbf{x}_G^T (\mathbf{V} - \mathbf{E}) \mathbf{x}_G + 2\bar{\lambda}_1 \|\bar{\mathbf{x}} - \mathbf{x}_G\|_2 - \hat{\mathbf{v}}^T (\mathbf{V} - \mathbf{E}) \hat{\mathbf{v}} \tag{64}$$

$$\leq \hat{\mathbf{v}}^T \mathbf{E} \hat{\mathbf{v}} - \mathbf{x}_G^T \mathbf{E} \mathbf{x}_G + 2\bar{\lambda}_1 \|\bar{\mathbf{x}} - \mathbf{x}_G\|_2 \tag{65}$$

$$= \tilde{\mathbf{x}}^T \mathbf{E} \tilde{\mathbf{x}} + 2 \left( \frac{\hat{\mathbf{v}} - \tilde{\mathbf{x}}}{2} \right)^T \mathbf{E} \left( \frac{\hat{\mathbf{v}} + \tilde{\mathbf{x}}}{2} \right) + 2 \left( \frac{\hat{\mathbf{v}} + \tilde{\mathbf{x}}}{2} \right)^T \mathbf{E} \left( \frac{\hat{\mathbf{v}} - \tilde{\mathbf{x}}}{2} \right) - \mathbf{x}_G^T \mathbf{E} \mathbf{x}_G + 2\bar{\lambda}_1 \|\bar{\mathbf{x}} - \mathbf{x}_G\|_2 \tag{66}$$

$$\leq \tilde{\mathbf{x}}^T \mathbf{E} \tilde{\mathbf{x}} + 2\delta \|\mathbf{E}\|_{2 \to 2} - \mathbf{x}_G^T \mathbf{E} \mathbf{x}_G + 2\bar{\lambda}_1 \|\bar{\mathbf{x}} - \mathbf{x}_G\|_2 \tag{67}$$

$$= 2 \left( \frac{\tilde{\mathbf{x}} + \mathbf{x}_G}{2} \right)^T \mathbf{E} \left( \frac{\tilde{\mathbf{x}} - \mathbf{x}_G}{2} \right) + 2 \left( \frac{\tilde{\mathbf{x}} - \mathbf{x}_G}{2} \right)^T \mathbf{E} \left( \frac{\tilde{\mathbf{x}} + \mathbf{x}_G}{2} \right) + 2\delta \|\mathbf{E}\|_{2 \to 2} + 2\bar{\lambda}_1 \|\bar{\mathbf{x}} - \mathbf{x}_G\|_2 \tag{68}$$

$$\leq 2C \sqrt{\frac{k \log \frac{4Lr}{\delta}}{m}} \cdot \|\tilde{\mathbf{x}} - \mathbf{x}_G\|_2 + 2\delta \|\mathbf{E}\|_{2 \to 2} + 2\bar{\lambda}_1 \|\bar{\mathbf{x}} - \mathbf{x}_G\|_2 \tag{69}$$

$$\leq 2C \sqrt{\frac{k \log \frac{4Lr}{\delta}}{m}} \cdot \left( \|\tilde{\mathbf{x}} - \hat{\mathbf{v}}\|_2 + \|\hat{\mathbf{v}} - \bar{\mathbf{x}}\|_2 + \|\bar{\mathbf{x}} - \mathbf{x}_G\|_2 \right) + 2\delta \|\mathbf{E}\|_{2 \to 2} + 2\bar{\lambda}_1 \|\bar{\mathbf{x}} - \mathbf{x}_G\|_2 \tag{70}$$

$$\leq 2C \sqrt{\frac{k \log \frac{4Lr}{\delta}}{m}} \cdot \|\hat{\mathbf{v}} - \bar{\mathbf{x}}\|_2 + O\left( \frac{\delta n}{m} \right) + O\left( (\bar{\lambda}_1 + \epsilon_n) \|\bar{\mathbf{x}} - \mathbf{x}_G\|_2 \right), \tag{71}$$

where:

- (57)–(58) follow from $\|\bar{\mathbf{x}}\|_2 = \|\hat{\mathbf{v}}\|_2 = 1$ and hence $|\bar{\mathbf{x}}^T \hat{\mathbf{v}}| \leq 1$;

- (60) follows since $(\bar{\lambda}_1, \bar{\mathbf{x}})$ are an eigenvalue-eigenvector pair for $\bar{\mathbf{V}}$ with $\|\bar{\mathbf{x}}\|_2 = 1$;

- (61) follows from (50);

- (62) follows from (53) with $\bar{\mathbf{V}}$ being symmetric and setting $\mathbf{s}_1 = \bar{\mathbf{x}}$, $\mathbf{s}_2 = \mathbf{x}_G$;

- (63) follows from $\|\bar{\mathbf{x}} + \mathbf{x}_G\|_2 \leq 2$ and $\|\bar{\mathbf{V}}\|_{2 \to 2} = \bar{\lambda}_1$;

- (64) follows since $\bar{\mathbf{V}} = \mathbf{V} - \mathbf{E}$;

- (65) follows since $\hat{\mathbf{v}}$ is a globally optimal solution to (5) and $\mathbf{x}_G \in \mathrm{Range}(G)$;

- (66) follows from (52) with $\mathbf{s}_1 = \hat{\mathbf{v}}$ and $\mathbf{s}_2 = \tilde{\mathbf{x}}$;

- (67) follows from (55) along with $\|\hat{\mathbf{v}} - \bar{\mathbf{x}}\|_2 \leq \delta$ and $\|\hat{\mathbf{v}} + \tilde{\mathbf{x}}\|_2 \leq 2$;

- (68) follows from (52);

- (69) follows from Assumption 2 (with $S_1 = S_2$ being $G(M)$ shifted by $\mathbf{x}_G$) and (54);

- (70) follows from the triangle inequality;

- (71) follows by substituting $\|\hat{\mathbf{v}} - \tilde{\mathbf{x}}\|_2 \leq \delta$, along with the assumptions $\|\mathbf{E}\|_{2 \to 2} = O(n/m)$, $m = \Omega\left( k \log \frac{Lr}{\delta} \right)$, and $\epsilon_n = O\left( \sqrt{\frac{k \log \frac{Lr}{\delta}}{m}} \right)$.

From (71), we have the following when $\hat{\mathbf{v}}^T \bar{\mathbf{x}} \geq 0$:

$$\|\hat{\mathbf{v}} - \bar{\mathbf{x}}\|_2 = \frac{O\left( \sqrt{\frac{k \log \frac{Lr}{\delta}}{m}} \right)}{\bar{\lambda}_1 - \bar{\lambda}_2} + O\left( \sqrt{\frac{\delta n/m}{\bar{\lambda}_1 - \bar{\lambda}_2}} \right) + O\left( \sqrt{\frac{(\bar{\lambda}_1 + \epsilon_n) \|\bar{\mathbf{x}} - \mathbf{x}_G\|_2}{\bar{\lambda}_1 - \bar{\lambda}_2}} \right). \tag{72}$$

As mentioned earlier, if $\hat{\mathbf{v}}^T\bar{\mathbf{x}} < 0$, we have the same upper bound as in (72) for $\|\hat{\mathbf{v}}+\bar{\mathbf{x}}\|_2$. Therefore, we obtain

$$\|\hat{\mathbf{v}}\hat{\mathbf{v}}^T - \bar{\mathbf{x}}\bar{\mathbf{x}}^T\|_{\mathrm{F}} = \sqrt{2\left(1 - (\bar{\mathbf{x}}^T\hat{\mathbf{v}})^2\right)} \tag{73}$$

$$= \sqrt{2\left(1 - \bar{\mathbf{x}}^T\hat{\mathbf{v}}\right)\left(1 + \bar{\mathbf{x}}^T\hat{\mathbf{v}}\right)} \tag{74}$$

$$\leq \sqrt{2}\min\{\|\hat{\mathbf{v}} - \bar{\mathbf{x}}\|_2, \|\hat{\mathbf{v}} + \bar{\mathbf{x}}\|_2\} \tag{75}$$

$$= \frac{O\left(\sqrt{\frac{k\log\frac{Lr}{\delta}}{m}}\right)}{\bar{\lambda}_1 - \bar{\lambda}_2} + O\left(\sqrt{\frac{\delta n/m}{\bar{\lambda}_1 - \bar{\lambda}_2}}\right) + O\left(\sqrt{\frac{(\bar{\lambda}_1 + \epsilon_n)\|\bar{\mathbf{x}} - \mathbf{x}_G\|_2}{\bar{\lambda}_1 - \bar{\lambda}_2}}\right), \tag{76}$$

where (73) follows from (40), (75) follows from $\|\hat{\mathbf{v}} \pm \bar{\mathbf{x}}\|_2^2 = 2(1 \pm \bar{\mathbf{x}}^T\hat{\mathbf{v}})$, and (76) follows from (72).

## E  PROOF OF LEMMA 2 (AUXILIARY RESULT FOR PPOWER ANALYSIS)

By the assumption $\mathrm{Range}(G) \subseteq \mathcal{S}^{n-1}$, for any $\mathbf{x} \in \mathbb{R}^n$ and $a > 0$, we have

$$\mathcal{P}_G(\mathbf{x}) = \mathcal{P}_G(a\mathbf{x}), \tag{77}$$

which is seen by noting that when comparing $\|\mathbf{x}-\mathbf{a}\|_2$ with $\|\mathbf{x}-\mathbf{b}\|_2$ (in accordance with projection mapping to the closest point), as long as $\|\mathbf{a}\|_2 = \|\mathbf{b}\|_2$, the comparison reduces to comparing $\langle\mathbf{x}, \mathbf{a}\rangle$ with $\langle\mathbf{x}, \mathbf{b}\rangle$, so is invariant to positive scaling of $\mathbf{x}$.

Let $\bar{\eta} = 1/\bar{\lambda}_1 > 0$ and $\hat{\mathbf{s}} = \mathcal{P}_G(\mathbf{V}\mathbf{s})$. Then, we have $\hat{\mathbf{s}} = \mathcal{P}_G(\mathbf{V}\mathbf{s}) = \mathcal{P}_G(\bar{\eta}\mathbf{V}\mathbf{s})$. Since $\bar{\mathbf{x}} \in \mathrm{Range}(G)$, we have

$$\|\bar{\eta}\mathbf{V}\mathbf{s} - \hat{\mathbf{s}}\|_2 \leq \|\bar{\eta}\mathbf{V}\mathbf{s} - \bar{\mathbf{x}}\|_2. \tag{78}$$

This is equivalent to

$$\|(\bar{\eta}\mathbf{V}\mathbf{s} - \bar{\mathbf{x}}) + (\bar{\mathbf{x}} - \hat{\mathbf{s}})\|_2^2 \leq \|\bar{\eta}\mathbf{V}\mathbf{s} - \bar{\mathbf{x}}\|_2^2, \tag{79}$$

and expanding the square gives

$$\|\bar{\mathbf{x}} - \hat{\mathbf{s}}\|_2^2 \leq 2\langle\bar{\eta}\mathbf{V}\mathbf{s} - \bar{\mathbf{x}}, \hat{\mathbf{s}} - \bar{\mathbf{x}}\rangle. \tag{80}$$

Note also that from $\bar{\mathbf{V}}\bar{\mathbf{x}} = \bar{\lambda}_1\bar{\mathbf{x}}$, we obtain $\bar{\mathbf{x}} = \bar{\eta}\bar{\mathbf{V}}\bar{\mathbf{x}}$, which we will use throughout the proof.

For $\delta > 0$, let $M$ be a $(\delta/L)$-net of $B_2^k(r)$; from Lemma 5, there exists such a net with

$$\log|M| \leq k\log\frac{4Lr}{\delta}. \tag{81}$$

By the $L$-Lipschitz continuity of $G$, we have that $G(M)$ is a $\delta$-net of $\mathrm{Range}(G) = G(B_2^k(r))$. We write

$$\mathbf{s} = (\mathbf{s} - \mathbf{s}_0) + \mathbf{s}_0, \quad \hat{\mathbf{s}} = (\hat{\mathbf{s}} - \tilde{\mathbf{s}}) + \tilde{\mathbf{s}}, \tag{82}$$

where $\tilde{\mathbf{s}} \in G(M)$ satisfies $\|\hat{\mathbf{s}} - \tilde{\mathbf{s}}\|_2 \leq \delta$, and $\mathbf{s}_0 \in G(M)$ satisfies $\|\mathbf{s} - \mathbf{s}_0\|_2 \leq \delta$. Then, we have

$$\langle\bar{\eta}\mathbf{V}\mathbf{s} - \bar{\mathbf{x}}, \hat{\mathbf{s}} - \bar{\mathbf{x}}\rangle = \langle\bar{\eta}\bar{\mathbf{V}}(\mathbf{s} - \bar{\mathbf{x}}), \hat{\mathbf{s}} - \bar{\mathbf{x}}\rangle + \langle\bar{\eta}\mathbf{E}\mathbf{s}, \hat{\mathbf{s}} - \bar{\mathbf{x}}\rangle, \tag{83}$$

which follows from $\mathbf{V} = \bar{\mathbf{V}} + \mathbf{E}$ and $\bar{\mathbf{x}} = \bar{\eta}\bar{\mathbf{V}}\bar{\mathbf{x}}$. In the following, we control the two terms in (83) separately.

1. The term $\langle\bar{\eta}\bar{\mathbf{V}}(\mathbf{s}-\bar{\mathbf{x}}), \hat{\mathbf{s}} - \bar{\mathbf{x}}\rangle$: We decompose $\mathbf{s} = \alpha\bar{\mathbf{x}} + \beta\mathbf{t}$ and $\hat{\mathbf{s}} = \hat{\alpha}\bar{\mathbf{x}} + \hat{\beta}\hat{\mathbf{t}}$, where $\|\mathbf{t}\|_2 = \|\hat{\mathbf{t}}\|_2 = 1$ and $\mathbf{t}^T\bar{\mathbf{x}} = \hat{\mathbf{t}}^T\bar{\mathbf{x}} = 0$. Since $\|\mathbf{s}\|_2 = \|\hat{\mathbf{s}}\|_2 = 1$, we have $\alpha^2 + \beta^2 = \hat{\alpha}^2 + \hat{\beta}^2 = 1$. In addition, we have $\alpha = \mathbf{s}^T\bar{\mathbf{x}}$ and $\hat{\alpha} = \hat{\mathbf{s}}^T\bar{\mathbf{x}}$. Recall that in (47), we write the SVD of $\bar{\mathbf{V}}$ as $\bar{\mathbf{V}} = \bar{\mathbf{U}}\bar{\mathbf{D}}\bar{\mathbf{U}}^T$. Since $\mathbf{t}^T\bar{\mathbf{x}} = 0$, we can write $\mathbf{t}$ as $\mathbf{t} = \sum_{i>1} h_i\bar{\mathbf{u}}_i$. In addition, since

$\|\mathbf{t}\|_2 = 1$, we have $\sum_{i>1} h_i^2 = 1$. Hence, by the Cauchy-Schwarz inequality, we have

$$|\langle \bar{\mathbf{V}}\mathbf{t}, \hat{\mathbf{t}} \rangle| \leq \|\bar{\mathbf{V}}\mathbf{t}\|_2 \tag{84}$$

$$= \left\| \sum_{i>1} \bar{\lambda}_i h_i \bar{\mathbf{u}}_i \right\|_2 \tag{85}$$

$$= \sqrt{\sum_{i>1} \bar{\lambda}_i^2 h_i^2} \tag{86}$$

$$\leq \sqrt{\bar{\lambda}_2^2 \sum_{i>1} h_i^2} \tag{87}$$

$$= \bar{\lambda}_2. \tag{88}$$

Therefore, we obtain

$$|\langle \bar{\eta}\bar{\mathbf{V}}(\mathbf{s} - \bar{\mathbf{x}}), \hat{\mathbf{s}} - \bar{\mathbf{x}} \rangle| = |\langle (\alpha - 1)\bar{\mathbf{x}} + \bar{\eta}\beta\bar{\mathbf{V}}\mathbf{t}, (\hat{\alpha} - 1)\bar{\mathbf{x}} + \hat{\beta}\hat{\mathbf{t}} \rangle| \tag{89}$$

$$= |(\alpha - 1)(\hat{\alpha} - 1) + \bar{\eta}\beta\hat{\beta}\langle \bar{\mathbf{V}}\mathbf{t}, \hat{\mathbf{t}} \rangle| \tag{90}$$

$$\leq (1 - \alpha)(1 - \hat{\alpha}) + \bar{\eta}|\beta\hat{\beta}|\bar{\lambda}_2 \tag{91}$$

$$= (1 - \alpha)(1 - \hat{\alpha}) + \bar{\gamma}\sqrt{1 - \alpha^2}\sqrt{1 - \hat{\alpha}^2}, \tag{92}$$

where (89) uses $\eta\bar{\mathbf{V}}\bar{\mathbf{x}} = \bar{\mathbf{x}}$, and (90) uses $\|\bar{\mathbf{x}}\|_2 = 1$ and $\langle \bar{\mathbf{x}}, \mathbf{t} \rangle = 0$.

2. The term $\langle \bar{\eta}\mathbf{E}\mathbf{s}, \hat{\mathbf{s}} - \bar{\mathbf{x}} \rangle$: We have

$$|\langle \bar{\eta}\mathbf{E}\mathbf{s}, \hat{\mathbf{s}} - \bar{\mathbf{x}} \rangle| = \langle \bar{\eta}\mathbf{E}((\mathbf{s} - \mathbf{s}_0) + \mathbf{s}_0), \hat{\mathbf{s}} - \bar{\mathbf{x}} \rangle \tag{93}$$

$$= \langle \bar{\eta}\mathbf{E}(\mathbf{s} - \mathbf{s}_0), \hat{\mathbf{s}} - \bar{\mathbf{x}} \rangle + \langle \bar{\eta}\mathbf{E}\mathbf{s}_0, (\hat{\mathbf{s}} - \tilde{\mathbf{s}}) + (\tilde{\mathbf{s}} - \bar{\mathbf{x}}) \rangle \tag{94}$$

$$\leq \bar{\eta}\|\mathbf{E}\|_{2\to 2}\delta\|\hat{\mathbf{s}} - \bar{\mathbf{x}}\|_2 + \bar{\eta}\|\mathbf{E}\|_{2\to 2}\delta + O\left(\sqrt{\frac{k \log \frac{Lr}{\delta}}{m}}\right) \cdot \|\tilde{\mathbf{s}} - \bar{\mathbf{x}}\|_2 \tag{95}$$

$$\leq O\left(\delta\|\mathbf{E}\|_{2\to 2}\right) + O\left(\sqrt{\frac{k \log \frac{Lr}{\delta}}{m}}\right) \cdot \|\hat{\mathbf{s}} - \bar{\mathbf{x}}\|_2, \tag{96}$$

where (95) follows from Assumption 2 and (81), and (96) follows from $\bar{\eta} = 1/\bar{\lambda}_1$, along with the fact that we assumed $\bar{\lambda}_1 = \Theta(1)$.

Note that $\|\bar{\mathbf{x}} - \hat{\mathbf{s}}\|_2^2 = 2(1 - \hat{\mathbf{s}}^T \bar{\mathbf{x}}) = 2(1 - \hat{\alpha})$. Hence, and using (80), (83), (92), and (96), we obtain

$$2(1 - \hat{\alpha}) \leq 2\left((1 - \alpha)(1 - \hat{\alpha}) + \bar{\gamma}\sqrt{1 - \alpha^2}\sqrt{1 - \hat{\alpha}^2}\right)$$

$$+ O\left(\delta\|\mathbf{E}\|_{2\to 2}\right) + O\left(\sqrt{\frac{k \log \frac{Lr}{\delta}}{m}}\right) \cdot \sqrt{2(1 - \hat{\alpha})}. \tag{97}$$

Using $2(1 - \hat{\alpha}) - 2(1 - \alpha)(1 - \hat{\alpha}) = 2\alpha(1 - \hat{\alpha})$, $\sqrt{1 - \alpha^2} = \sqrt{1 - \alpha}\sqrt{1 + \alpha} \leq \sqrt{2(1 - \alpha)}$, and similarly $\sqrt{1 - \hat{\alpha}^2} \leq \sqrt{2(1 - \hat{\alpha})}$, we obtain from (97) that

$$2\alpha(1 - \hat{\alpha}) \leq 2\bar{\gamma}\sqrt{2(1 - \alpha)}\sqrt{2(1 - \hat{\alpha})} + O\left(\sqrt{\frac{k \log \frac{Lr}{\delta}}{m}}\right) \cdot \sqrt{2(1 - \hat{\alpha})} + O\left(\delta\|\mathbf{E}\|_{2\to 2}\right). \tag{98}$$

Since $\|\hat{\mathbf{s}} - \bar{\mathbf{x}}\|_2^2 = 2(1 - \hat{\alpha})$ and $\|\mathbf{s} - \bar{\mathbf{x}}\|_2^2 = 2(1 - \alpha)$, this is equivalent to

$$\alpha\|\hat{\mathbf{s}} - \bar{\mathbf{x}}\|_2^2 \leq \left(2\bar{\gamma}\|\mathbf{s} - \bar{\mathbf{x}}\|_2 + O\left(\sqrt{\frac{k \log \frac{Lr}{\delta}}{m}}\right)\right)\|\hat{\mathbf{s}} - \bar{\mathbf{x}}\|_2 + O\left(\delta\|\mathbf{E}\|_{2\to 2}\right). \tag{99}$$

This equation is of the form $az^2 \leq bz + c$ (where $z = \|\hat{\mathbf{s}} - \bar{\mathbf{x}}\|_2$ and $a = \alpha == \mathbf{s}^T\bar{\mathbf{x}} > 0$), and using a simple application of the quadratic formula,[9] we obtain

$$\|\hat{\mathbf{s}} - \bar{\mathbf{x}}\|_2 \leq \frac{2\bar{\gamma}\|\mathbf{s} - \bar{\mathbf{x}}\|_2}{\mathbf{s}^T\bar{\mathbf{x}}} + O\left(\frac{1}{\mathbf{s}^T\bar{\mathbf{x}}}\left(\sqrt{\frac{k\log\frac{Lr}{\delta}}{m}} + \sqrt{\mathbf{s}^T\bar{\mathbf{x}}\cdot\delta\|\mathbf{E}\|_{2\to 2}}\right)\right) \tag{100}$$

$$\leq \frac{2\bar{\gamma}\|\mathbf{s} - \bar{\mathbf{x}}\|_2}{\mathbf{s}^T\bar{\mathbf{x}}} + O\left(\frac{1}{\mathbf{s}^T\bar{\mathbf{x}}}\sqrt{\frac{k\log(nLr)}{m}}\right), \tag{101}$$

where we use the assumption $\|\mathbf{E}\|_{2\to 2} = O(n/m)$ and set $\delta = 1/n$ in (101).

## F  PROOF OF THEOREM 2 (MAIN THEOREM FOR PPOWER)

Suppose for the time being that (13) holds for at least one index $t \geq t_0$ (we will later verify that this is the case), and let $T_0 \geq t_0$ be the smallest such index. Thus, we have

$$\|\mathbf{w}^{(T_0)} - \bar{\mathbf{x}}\|_2 \leq \frac{C}{(1-\mu_0)\nu}\sqrt{\frac{k\log(nLr)}{m}}. \tag{102}$$

Note that according to the theorem statement, $1 - \mu_0$ is bounded away from zero. Using $\|\mathbf{w}^{(T_0)}\|_2 = \|\bar{\mathbf{x}}\|_2 = 1$ and the assumption that $m \geq C_{\nu,\tau}\cdot k\log(nLr)$ with $C_{\nu,\tau} > 0$ being large enough, we deduce from (102) that $\|\mathbf{w}^{(T_0)} - \bar{\mathbf{x}}\|_2$ is sufficiently small such that

$$\bar{\mathbf{x}}^T\mathbf{w}^{(T_0)} \geq 1 - \tau. \tag{103}$$

Next, using the assumption $2\bar{\gamma} + \nu \leq 1 - \tau$, we write

$$\frac{2\bar{\gamma}}{(1-\mu_0)\nu(1-\tau)} + \frac{1}{1-\tau} = \frac{2\bar{\gamma} + (1-\mu_0)\nu}{(1-\mu_0)\nu(1-\tau)} \leq \frac{2\bar{\gamma} + \nu}{(1-\mu_0)\nu(1-\tau)} \leq \frac{1}{(1-\mu_0)\nu}. \tag{104}$$

Then, from Lemma 2, we obtain

$$\|\mathbf{w}^{(T_0+1)} - \bar{\mathbf{x}}\|_2 \leq \frac{2\bar{\gamma}}{\bar{\mathbf{x}}^T\mathbf{w}^{(T_0)}}\cdot\|\mathbf{w}^{(T_0)} - \bar{\mathbf{x}}\|_2 + \frac{C}{\bar{\mathbf{x}}^T\mathbf{w}^{(T_0)}}\sqrt{\frac{k\log(nLr)}{m}} \tag{105}$$

$$\leq \frac{2\bar{\gamma}}{1-\tau}\cdot\frac{C}{(1-\mu_0)\nu}\sqrt{\frac{k\log(nLr)}{m}} + \frac{C}{1-\tau}\sqrt{\frac{k\log(nLr)}{m}} \tag{106}$$

$$\leq \frac{C}{(1-\mu_0)\nu}\sqrt{\frac{k\log(nLr)}{m}}, \tag{107}$$

where (106) follows from (102)–(103), and (107) follows from (104). Thus, we have transferred (102) from $T_0$ to $T_0 + 1$, and proceeding by induction, we obtain

$$\|\mathbf{w}^{(t)} - \bar{\mathbf{x}}\|_2 \leq \frac{C}{(1-\mu_0)\nu}\sqrt{\frac{k\log(nLr)}{m}} \tag{108}$$

for al $t \geq T_0$.

Next, we consider $t \in [t_0, T_0)$. Again using Lemma 2 (with $\hat{\mathbf{s}} = \mathbf{w}^{(t_0+1)} = \mathcal{P}_G(\mathbf{V}\mathbf{w}^{(t_0)})$), we have

$$\|\mathbf{w}^{(t_0+1)} - \bar{\mathbf{x}}\|_2 \leq \mu_0\|\mathbf{w}^{(t_0)} - \bar{\mathbf{x}}\|_2 + \frac{C}{2\bar{\gamma} + \nu}\cdot\sqrt{\frac{k\log(nLr)}{m}}, \tag{109}$$

where we recall that $\mu_0 = \frac{2\bar{\gamma}}{\bar{\mathbf{x}}^T\mathbf{w}^{(t_0)}} = \frac{2\bar{\gamma}}{2\bar{\gamma}+\nu} < 1$, and note that the denominator in the second term of (109) follows since $\bar{\mathbf{x}}^T\mathbf{w}^{(t_0)} = 2\bar{\gamma} + \nu$. Supposing that $t_0 < T_0$ (otherwise, the above analysis for $t \geq T_0$ alone is sufficient), we have that (13) is reversed at $t = t_0$:

$$\|\mathbf{w}^{(t_0)} - \bar{\mathbf{x}}\|_2 > \frac{C}{(1-\mu_0)\nu}\sqrt{\frac{k\log(nLr)}{m}}. \tag{110}$$

---

[9]Since the leading coefficient $a = \alpha$ of the quadratic is positive, $z$ must lie in between the two associated roots. This yields $z \leq \frac{b+\sqrt{b^2+4ac}}{2a}$, from which the inequality $\sqrt{a+b} \leq \sqrt{a} + \sqrt{b}$ gives (100).

This means that we can upper bound the second term in (109) by $\frac{C}{\nu} \cdot \sqrt{\frac{k \log(nLr)}{m}} < (1-\mu_0) \|\mathbf{w}^{(t_0)} - \bar{\mathbf{x}}\|_2$, which gives

$$\|\mathbf{w}^{(t_0+1)} - \bar{\mathbf{x}}\|_2 < \|\mathbf{w}^{(t_0)} - \bar{\mathbf{x}}\|_2. \tag{111}$$

Squaring both sides, expanding, and canceling the norms (which all equal one), we obtain

$$\bar{\mathbf{x}}^T \mathbf{w}^{(t_0+1)} > \bar{\mathbf{x}}^T \mathbf{w}^{(t_0)}, \tag{112}$$

and by induction, we obtain that $\{\bar{\mathbf{x}}^T \mathbf{w}^{(t)}\}_{t \in [t_0, T_0)}$ is monotonically increasing.

Recall that we assume that $\bar{\lambda}_1 = \Theta(1)$, and that $T_0 = t_0 + \Delta_0$ is the smallest integer such that (13) holds. To verify that $T_0$ is finite and upper bound $\Delta_0$, we consider the following three cases:

- $\mu_0 = 0$ (or equivalently, $\bar{\gamma} = \bar{\lambda}_2 = 0$): In this case, (109) gives $T_0 = t_0 + 1$ (or $T_0 = t_0$, which we already addressed above). Thus, we have $\Delta_0 \le 1$, as stated in the theorem.

- $\mu_0 = o(1)$ (or equivalently, $\bar{\gamma} = o(1)$ and $\bar{\lambda}_2 = o(1)$): Since $\{\bar{\mathbf{x}}^T \mathbf{w}^{(t)}\}_{t \in [t_0, T_0)}$ is monotonically increasing, for any positive integer $\Delta$ with $t_0 + \Delta \le T_0$, by applying Lemma 2 (or (109)) multiple times, we obtain[10]

$$\|\mathbf{w}^{(t_0+\Delta)} - \bar{\mathbf{x}}\|_2 \le \mu_0^\Delta \|\mathbf{w}^{(t_0)} - \bar{\mathbf{x}}\|_2 + \frac{1 - \mu_0^\Delta}{1 - \mu_0} \cdot \frac{C}{2\bar{\gamma} + \nu} \cdot \sqrt{\frac{k \log(nLr)}{m}} \tag{113}$$

$$\le \mu_0^\Delta \|\mathbf{w}^{(t_0)} - \bar{\mathbf{x}}\|_2 + \frac{1}{1 - \mu_0} \cdot \frac{C}{2\bar{\gamma} + \nu} \cdot \sqrt{\frac{k \log(nLr)}{m}}. \tag{114}$$

Then, if we choose $\Delta_0 \in \mathbb{N}$ such that

$$\mu_0^{\Delta_0 - 1} \le \frac{C}{2\nu} \cdot \sqrt{\frac{k \log(nLr)}{m}}, \tag{115}$$

we obtain from (114) that

$$\|\mathbf{w}^{(t_0+\Delta_0)} - \bar{\mathbf{x}}\|_2 \le \mu_0^{\Delta_0} \|\mathbf{w}^{(t_0)} - \bar{\mathbf{x}}\|_2 + \frac{1}{1 - \mu_0} \cdot \frac{C}{2\bar{\gamma} + \nu} \cdot \sqrt{\frac{k \log(nLr)}{m}} \tag{116}$$

$$\le 2\mu_0^{\Delta_0} + \frac{1}{1 - \mu_0} \cdot \frac{C}{2\bar{\gamma} + \nu} \cdot \sqrt{\frac{k \log(nLr)}{m}} \tag{117}$$

$$< \frac{C\mu_0}{\nu(1 - \mu_0)} \cdot \sqrt{\frac{k \log(nLr)}{m}} + \frac{1}{1 - \mu_0} \cdot \frac{C}{2\bar{\gamma} + \nu} \cdot \sqrt{\frac{k \log(nLr)}{m}} \tag{118}$$

$$= \frac{C}{(1 - \mu_0)\nu} \cdot \sqrt{\frac{k \log(nLr)}{m}}, \tag{119}$$

where (117) follows from $\|\mathbf{w}^{(t_0)} - \bar{\mathbf{x}}\|_2 \le 2$, (118) follows from (115) and $1 < \frac{1}{1-\mu_0}$, and (119) follows from $\mu_0 = \frac{2\bar{\gamma}}{2\bar{\gamma}+\nu}$, which implies $\frac{\mu_0}{\nu} + \frac{1}{2\bar{\gamma}+\nu} = \frac{\mu_0(2\bar{\gamma}+\nu)+\nu}{\nu(2\bar{\gamma}+\nu)} = \frac{2\bar{\gamma}+\nu}{\nu(2\bar{\gamma}+\nu)} = \frac{1}{\nu}$. Observe that (119) coincides with (13), and since $\mu_0 = o(1)$, we obtain from (115) that $\Delta_0 = O\left(\log\left(\frac{m}{k \log(nLr)}\right)\right)$ as desired.

- $\mu_0 = \Theta(1)$ (or equivalently, $\bar{\gamma} = \Theta(1)$ and $\bar{\lambda}_2 = \Theta(1)$): Recall that we only need to focus on the case $T_0 > t_0$. This means that (110) holds, implying that we can upper bound the second term in (109) by $\frac{(1-\mu_0)\nu}{2\bar{\gamma}+\nu} \cdot \|\mathbf{w}^{(t_0)} - \bar{\mathbf{x}}\|_2$, yielding

$$\|\mathbf{w}^{(t_0+1)} - \bar{\mathbf{x}}\|_2 < \mu_0 \|\mathbf{w}^{(t_0)} - \bar{\mathbf{x}}\|_2 + \frac{(1-\mu_0)\nu}{2\bar{\gamma} + \nu} \cdot \|\mathbf{w}^{(t_0)} - \bar{\mathbf{x}}\|_2 \tag{120}$$

$$= \frac{2\bar{\gamma} + (1-\mu_0)\nu}{2\bar{\gamma} + \nu} \cdot \|\mathbf{w}^{(t_0)} - \bar{\mathbf{x}}\|_2 \tag{121}$$

$$= (1 - \xi)\|\mathbf{w}^{(t_0)} - \bar{\mathbf{x}}\|_2, \tag{122}$$

---

[10] In simpler notation, if $z_{t+1} \le a z_t + b$, then we get $z_{t+2} \le a^2 z_t + (1+a)b$, then $z_{t+3} \le a^3 z_t + (1+a+a^2)b$, and so on, and then we can apply $1 + a + \ldots + a^{i-1} = \frac{1-a^i}{1-a}$ for $a \ne 1$.

where $\xi = \frac{\mu_0 \nu}{2\bar{\gamma}+\nu} = \mu_0(1-\mu_0) = \Theta(1)$. With the distance to $\bar{\mathbf{x}}$ shrinking by a constant factor in each iteration according to (122), and the initial distance $\|\mathbf{w}^{(t_0)} - \bar{\mathbf{x}}\|_2$ being at most 2 due to the vectors having unit norm, we deduce that $\Delta_0 = O\left(\log\left(\frac{m}{k \log(nLr)}\right)\right)$ iterations suffice to ensure that (13) holds for $t = t_0 + \Delta_0$.

## G  COMPARISON OF ANALYSIS TO DESHPANDE ET AL. (2014)

As mentioned in Section 4, our analysis is significantly different from that of Deshpande et al. (2014) despite using a similar assumption on the initialization. We highlight the differences as follows:

1. Perhaps the most significant difference is that the proof of (Deshpande et al., 2014, Theorem 3) is highly dependent on the Moreau decomposition, which is only valid for a closed convex cone (see (Deshpande et al., 2014, Definition 1.2)). In particular, the Moreau decomposition needs to be used at the beginning of the proof of (Deshpande et al., 2014, Theorem 3), such as Eqs. (18) and (19) in the supplementary material therein. We do not see a way for the proof to proceed without the Moreau decomposition, and our $\text{Range}(G)$ may be very different from a convex cone.

2. We highlight that one key observation in our proof of Lemma 2 (and thus Theorem 2) is that for a generative model $G$ with $\text{Range}(G) \subseteq \mathcal{S}^{n-1}$, and any $\mathbf{x} \in \mathbb{R}^n$ and $a > 0$, we have $\mathcal{P}_G(a\mathbf{x}) = \mathcal{P}_G(\mathbf{x})$ (Eq. (77)). This enables us to derive the important equation $\hat{\mathbf{s}} = \mathcal{P}_G(\mathbf{V}\mathbf{s}) = \mathcal{P}_G(\bar{\eta}\mathbf{V}\mathbf{s})$. We are not aware of a similar idea being used in the proof of (Deshpande et al., 2014, Theorem 3).

3. In the PPower method in (Deshpande et al., 2014), the authors need to add $\rho\mathbf{I}_n$ with $\rho > 0$ to the observed data matrix $\mathbf{V}$ to improve the convergence. In particular, they mention in the paragraph before the statement of Theorem 3 that "the memory term $\rho\mathbf{v}^t$ is necessary for our proof technique to go through". In contrast, our proof of Theorem 2 does not require adding such terms, even when our data model is restricted to the spiked Wigner model considered in (Deshpande et al., 2014).

4. We consider a matrix model that is significantly more general than the spiked Wigner model studied in (Deshpande et al., 2014).

## H  NUMERICAL RESULTS FOR THE FASHION-MNIST DATASET

The Fashion-MNIST dataset consists of Zalando's article images with a training set of $60,000$ examples and a test set of $10,000$ examples. The size of each image in the Fashion-MNIST dataset is also $28 \times 28$, and thus $n = 784$.

The generative model $G$ is set to be a boundary-seeking generative adversarial network (BEGAN). The BEGAN architecture is summarized as follows.[11] The generator has latent dimension $k = 62$ and four layers. The first two are fully connected layers with the architecture $62 - 1024 - 6272$, and with ReLU activation functions. The output of the second layer, reshaped to $128 \times 7 \times 7$, is forwarded to a deconvolution layer with kernel size 4 and stride 2. The third layer uses ReLU activations and has output size $64 \times 14 \times 14$, where 64 is the number of channels. The fourth layer is a deconvolution layer with kernel size 4 and strides 2, and it uses ReLU activations and has output size $1 \times 28 \times 28$, where the number of channels is 1.

The BEGAN is trained with a mini-batch size of 256, a learning rate of 0.0002, and 100 epochs. The other parameters are the same as those for the MNIST dataset. We perform two sets of experiments, considering the spiked covariance and phase retrieval models separately. The corresponding results are reported in Figures 4, 5, 6, and 7. From these figures, again, we can observe clear superiority of `PPower` to `Power` and `TPower`. We note that for the Fashion-MNIST dataset, some of the images are not sparse in the natural basis, but we observe from Figures 4 and 6 that even for the sparsest images (sandals), PPower also significantly outperforms TPower.

---

[11]Further details of the architecture can be found at `https://github.com/hwalsuklee/tensorflow-generative-model-collections`.

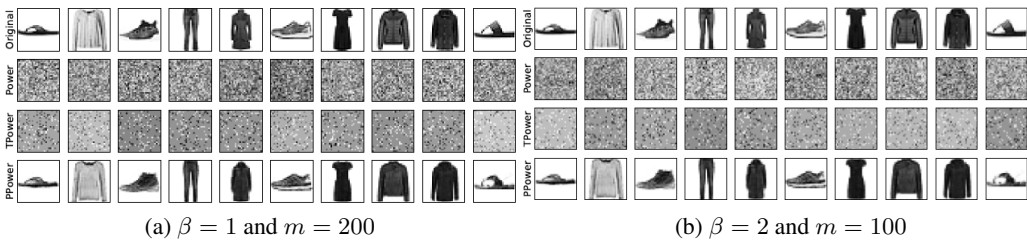

(a) $\beta = 1$ and $m = 200$        (b) $\beta = 2$ and $m = 100$

Figure 4: Examples of reconstructed Fashion-MNIST images for the spiked covariance model.

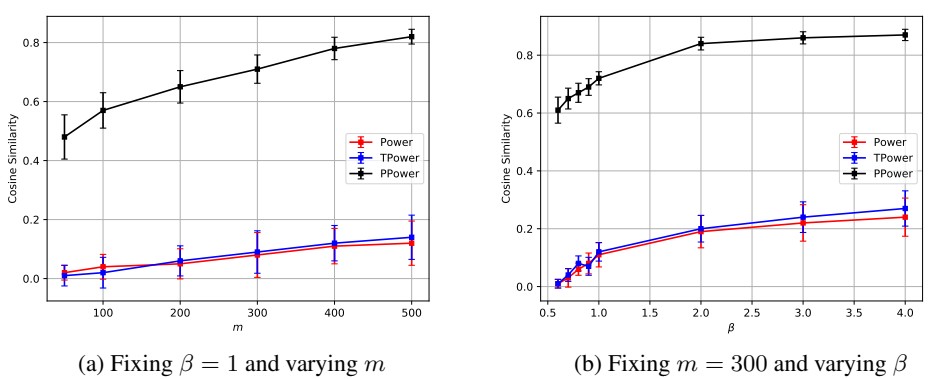

(a) Fixing $\beta = 1$ and varying $m$        (b) Fixing $m = 300$ and varying $\beta$

Figure 5: Quantitative comparisons of the performance of Power, TPower and PPower according to the Cosine Similarity for the Fashion-MNIST dataset and the spiked covariance model.

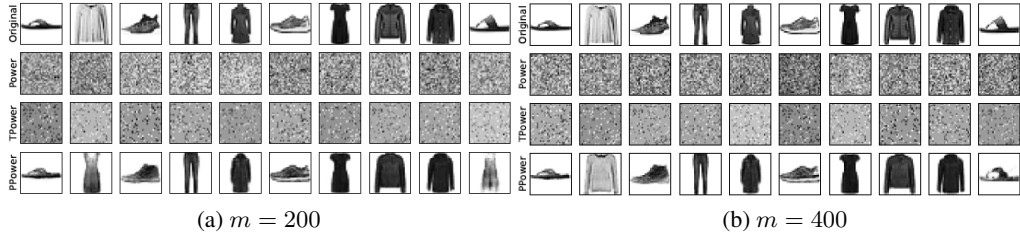

(a) $m = 200$        (b) $m = 400$

Figure 6: Examples of reconstructed images of the Fashion-MNIST dataset for phase retrieval.

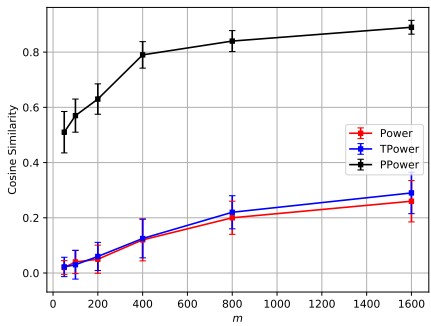

Figure 7: Quantitative comparisons of the performance of Power, TPower and PPower according to the Cosine Similarity for the Fashion-MNIST dataset and the phase retrieval model.

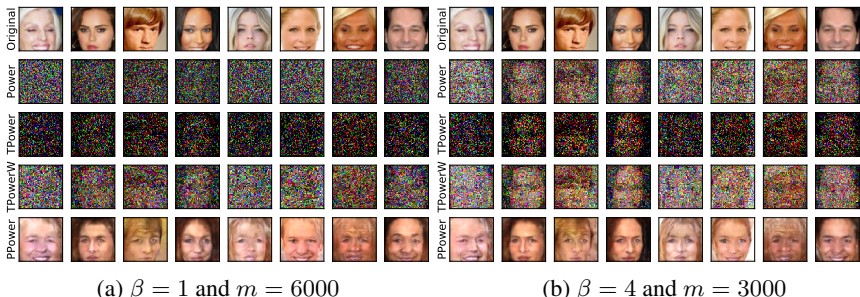

(a) $\beta = 1$ and $m = 6000$                    (b) $\beta = 4$ and $m = 3000$

Figure 8: Examples of reconstructed CelebA images for the spiked covariance model.

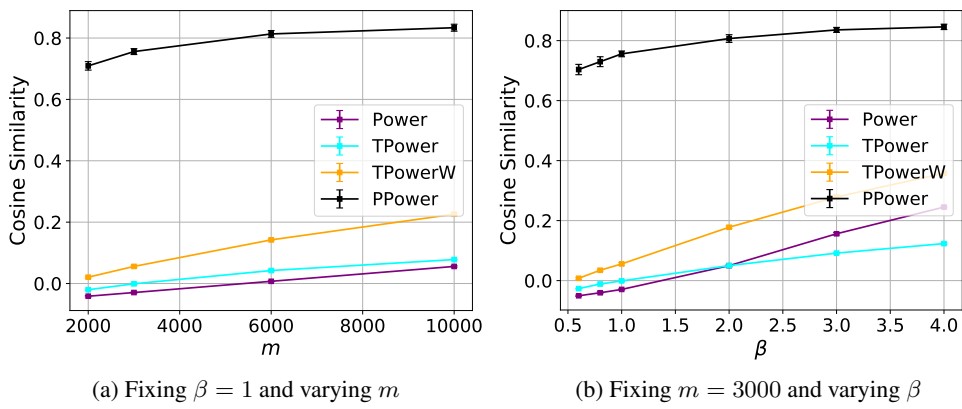

(a) Fixing $\beta = 1$ and varying $m$                    (b) Fixing $m = 3000$ and varying $\beta$

Figure 9: Quantitative comparisons of the performance of `Power`, `TPower`, `TPowerW` and `PPower` according to the Cosine Similarity for the CelebA dataset and spiked covariance model.

## I   NUMERICAL RESULTS FOR THE CELEBA DATASET

The CelebA dataset consists of more than $200,000$ face images of celebrities, where each input image is cropped to a $64 \times 64$ RGB image with $n = 64 \times 64 \times 3 = 12288$. The generative model $G$ is set to be a pre-trained Deep Convolutional Generative Adversarial Networks (DCGAN) model with latent dimension $k = 100$. We use the DCGAN model trained by the authors of (Bora et al., 2017) directly. We select the best estimate among 2 random restarts. The Adam optimizer with 100 steps and a learning rate of $0.1$ is used for the projection operator.

For the images of the CelebA dataset, the corresponding vectors are clearly not sparse in the natural basis. To make a fairer comparison to the sparsity-based method `TPower`, we convert the original images to the wavelet basis, and perform `TPower` on these converted images. The obtained results of `TPower` are then converted back to the vectors in the natural basis. The corresponding method is denoted by `TPowerW`, with "W" referring to the conversion to images in the wavelet basis. In each iteration of `TPower` and `TPowerW`, the calculated entries are truncated to zero except for the largest $q$ entries. For CelebA, $q$ is set to be $2000$. Other parameters are the same as those for the MNIST dataset. We also perform two sets of experiments, considering the spiked covariance and phase retrieval models separately. The corresponding results are reported in Figures 8 and 9. From these figures, we can observe the superiority of `PPower` to `Power`, `TPower` and `TPowerW`, whereas `TPowerW` only marginally improves over `TPower`.

