# OpenReview forum: "Generative Principal Component Analysis"
_ICLR.cc/2022/Conference — ICLR 2022 Poster_

### Official Review · Reviewer_kCBb · 2021-10-29

**Correctness:** 3
**Technical Novelty And Significance:** 4
**Empirical Novelty And Significance:** 3
**Recommendation:** 6
**Confidence:** 4

**Main Review:**

Strengths:

1) The problem is important and well-stated. The paper is clearly written. Theoretical and algorithmic developments that can move the needle away from sparsity-based assumptions are very much needed, and this paper can be an interesting development in that direction.

2) The assumptions and the theoretical results are plausible.

Weakness:

1) My biggest concern is theoretical, and a little bit philosophical. From what I can tell, the generative model $G$ is trained \textit{on the same data} for which the eigenvector computation is desired. This implies that the training of $G$ and its performance is not independent, and this undercuts the theoretical results presented in a major way. Additionally, this can result in the "Richard Feynman's license plate" phenomenon (see \url{https://www.quora.com/Richard-Feyman-had-once-remarked-about-an-amazing-experience-where-he-saw-a-car-with-the-license-plate-ARW-357-Did-it-have-any-significance?share=1} for a
pop-science version) or statistical superefficiency, which results in underestimation of uncertainty and false precision/accuracy in other contexts.

Also, it is not fair to compare PPower with G trained on the same data with other algorithms that do not use such additional information.

This issue can be taken care of by partitioning the data to ensure that the training of G and the PPower computations are independent, but that would be a major revision of this paper as it stands now.

[REVISED COMMENT:] Based on the author's response, I think my main concern with this paper (stated above) is addressed.

2) A second major issue is computational. How expensive is the pre-training of G followed by PPower compared to rival algorithms? My guess is that it would be substantially more expensive, since learning $G$ is a very expensive step.

3) Authors have done a reasonable review of the literature on sparse PCA, but missed those that try to balance statistical efficiency with computational complexity, see Wang et al, Annals of Statistics, 2016, pages 1896-1930 and related papers (there are papers before and after this one on this topic). I think the authors will find the PPower scheme is very high on statistical efficiency, which is a good thing, but requires detailed discussion.

Also, the above reference and related papers will give authors ideas about more efficient algorithms for the case when the leading eigenvector is actually sparse: the Power and TPower algorithms are essentially strawmen.


**Summary Of The Paper:**

Principal component analysis (PCA) is one of the most commonly used techniques for dimension reduction in data and for explaining variability of the data in terms of its eigenvalues/vectors. Unfortunately in high-dimensions, PCA is mathematically inconsistent: this is well-known from random matrix literature. Hence the standard approach happens to be to assume that there is sparsity in the leading eigenvectors. In this paper however, motivated by Bora et. al (2017), the study is based on the conjecture/premise that sparsity assumption may be replaceable by the assumption that the leading eigenvector is close to the range of a deep generative model.


**Summary Of The Review:**

The problem is important and the idea of replacing sparsity with a generative model is very interesting. However, technical concerns remain.


Revised comment: I think the authors' for their detailed and pertinent responses to comments. My major concerns are now addressed, and I am happy to assign an imporved score. I think this paper can be a major contribution, and I agree with the authors that some of the criticisms (for example, on computations) that can be leveled at this paper are relevant to the entire gamut of papers following Bora et al. Since there is no free lunch, I am willing to accept the argument that there must be a bit of give in terms of training cost or some toher tings to obtain this new kind of PCA.

---

> ### Author Response · Authors · 2021-11-11
> **Responses to Reviewer kCBb**
>
> Thanks for your useful comments. Concerning the three major concerns listed in your review (‘Weaknesses’ section), our responses are as follows.
>
> (**Concern 1 -- Training of G**)  We believe there has been a misunderstanding, and we have revised the paper accordingly; the details are given as follows.
>
> To train a VAE model for the MNIST dataset, Bora et al., (2017) split the dataset into training and test sets (e.g., in Section 6.1, they have mentioned that “In all cases, we report the results on a held out test set, unseen by the generative model at training time”).  For the MNIST dataset, in our experiments (and many other follow-up works of Bora et al., (2017), such as (Dhar et al., 2018; Liu et al., 2020)), we use the VAE model trained by Bora et al. directly. We also follow the setting in Bora et al., (2017) and only report the results on a test set that is unseen by the pre-trained VAE model. For Fashion-MNIST, we train a BEGAN following the codes shared by Bora et al. at https://github.com/AshishBora/csgm, and we have similarly split the data into training and test sets (e.g., in Appendix E in the submitted version, we state “The Fashion-MNIST dataset consists of Zalando’s article images with a training set of $60,000$ examples and a test set of $10,000$ examples”).
>
> To make as clear as possible that throughout our experiments, the training of $G$ and the PPower computations are independent, we have added the sentences “We use the VAE model trained by the authors of (Bora et al., 2017) directly.” and “Similarly to (Bora et al., 2017) and other related works, we only report the results on a test set that is unseen by the pre-trained VAE model, i.e., the training of G and the PPower computations do not use common data.” to our Experiments Section.
>
> Regarding the comparison to PPower, we are not aware of any existing baselines that utilise training data in this way, which is why no such baselines are included.  Basically, although the baselines may appear weak, they are the best that we are aware of from the existing literature.
>
> (**Concern 2 -- Computation for pre-training**) We follow the setting in Bora et al., (2017) to first train a generative model $G$ (or use an existing pre-trained $G$), and then perform the recovery task. We believe that for such a setting, the computation in training $G$ is of secondary importance, since (i) for popular datasets, pre-trained models are already readily available, and (ii) even when training needs to be done, many well-established training methods exist with well-documented computational requirements.  Finally, we remark on an analogy in dictionary learning, where it is similarly popular to learn a dictionary first, and then perform the subsequent recovery task based on the learned dictionary.
>
> To conclude the above discussions, we believe it is important to note that the reviewer’s first two major concerns are not unique to our paper, but also apply to the work of Bora et al., (2017) and the extensive follow-up works that it spawned, many of which have been highly influential and impactful. Overcoming these challenges is quite orthogonal to our main goal, which is adapting this line of works to a new problem (generative PCA).  We hope that the final decision is based on our main goal and contributions, rather than general limitations in a broad line of works.
>
> We have opted not to include the preceding discussion in the revised paper, but we would be happy to reconsider this decision on request.
>
> (**Concern 3 -- Missing reference**) Thanks for pointing out this interesting paper to us. We have cited it in the revised version, and can consider others if the reviewer has additional suggestions. We are aware of the computational-to-statistical gap in sparse PCA and sparse phase retrieval, and that such a gap has been mentioned in some works that are closely relevant to ours, e.g., ((Hand et al., 2018; Aubin et al., 2019; Cocola et al., 2020). We thank the reviewer for reminding us about this, and we have highlighted that our PPower algorithm partially closes the computational-to-statistical gap for both spiked matrix recovery and phase retrieval under a generative prior (replicated below).  We agree with the reviewer’s final statement that PPower enjoys good statistical efficiency, and we believe that our experiments also highlight its computational feasibility.
>
> The above-mentioned added sentence is: “We highlight that Theorem 2 partially closes the computational-to-statistical gap (e.g., see (Wang et al., 2016; Hand et al., 2018; Aubin et al., 2019; Cocola et al., 2020)) for spiked matrix recovery and phase retrieval under a generative prior, though closing it completely would require efficiently finding a good initialization and addressing the assumption of exact projections.” -- this can be found in Section 4.

---

> > ### Author Response · Authors · 2021-11-22
> > **Follow-up response to reviewer kCBb**
> >
> > Dear Reviewer kCBb,
> >
> > In addition to our initial response, we would like to corroborate our assertion that the first two major limitations mentioned (regarding the data and computation for pre-training $G$) are inherent to this line of works and orthogonal to our main goals in this submission, by highlighting some examples of existing papers where the same limitations apply. It is evident that these papers (among many others) are at top venues and/or highly cited, and we contend that this is sufficient evidence that they have been accepted by the community as an interesting line of ongoing investigation:
> >
> > [1] Bora et al. "Compressed sensing using generative models." ICML, 2017. [469 citations]
> >
> > [2] Mardani et al. "Deep generative adversarial neural networks for compressive sensing MRI." IEEE Trans. Medical Imaging, 2018. [298 citations]
> >
> > [3] Shah & Hegde. "Solving linear inverse problems using GAN priors: An algorithm with provable guarantees." ICASSP, 2018. [112 citations]
> >
> > [4] Dhar et al. "Modeling sparse deviations for compressed sensing using generative models." ICML, 2018. [46 citations]
> >
> > [5] Liu et al., “Sample complexity bounds for 1-bit compressive sensing and binary stable embeddings with generative priors.” ICML, 2020 [11 citations]
> >
> > [6] Jalal et al. “Robust compressed sensing MRI with deep generative priors”, NeurIPS, 2021 [new paper]
> >
> > [7] (Survey Article) Lucas et al. "Using deep neural networks for inverse problems in imaging: Beyond analytical methods." IEEE Sig. Proc. Magazine, 2018. [306 citations]
> >
> > [8] (Survey Article) Ongie et al. "Deep learning techniques for inverse problems in imaging." IEEE J. Selected Areas in Information Theory, 2020. [126 citations]

---

### Official Review · Reviewer_4eD5 · 2021-11-08

**Correctness:** 4
**Technical Novelty And Significance:** 3
**Empirical Novelty And Significance:** 3
**Recommendation:** 8
**Confidence:** 4

**Main Review:**

Strengths:
1. I think this paper does good work in its attempt to characterize both, and succeeds in characterizing the second **under a good initialization**.
2. Prior work either made assumptions on the type of GAN (such as being randomly initialized) or had algorithms that seemed not very practical. It is nice to see a potentially practical result in a more general setting, even if it relies on good initialization.

Weaknesses:
1. I am curious to see if GANs in practice satisfy lipschitzness, in particular the GAN that was used in the experiments. It would be nice to see some experiments to this effect.
2. I think the experiments are interesting and demonstrate (at least naively) that PPower is good. I would be interested in looking at the performance of TPower (the truncated power method, where you zero out all but the largest k entries) for these images in the *wavelet basis*, since that is where the method is more natural (since images are conjectured to be sparse in this basis).



**Summary Of The Paper:**

This paper studies the problem of PCA with a generative model setup. More precisely, they study the following setting
Given $\overline V = V + E$ where $V$ is PSD and $E$ is a perturbation matrix, find $\text{max}_w~ w^T V w$ where $w \in \text{Range}(G)$ where $G : B_2^k(r) \rightarrow \mathbb S^{n-1}$ is $L$-lipschitz.

The algorithm they analyze is a projected power method (PPower), similar to iterative hard thresholding for the case when $w$ is sparse. They demonstrate that if $w^{(i)} $ for $i \leq t$ are the $w$'s at each iteration, then for $Q(x) = x^T V x$, $Q(w^{(i)})$ is monotonically non-decreasing.

The subsequent results rely on the following two assumptions, which the authors demonstrate in the setting of the spiked wigner model, spiked covariance model and phase retrieval.

A. There is a gap between the largest and second largest eigenvalue of $\overline V$.
B. For $S_1, S_2 \subset \mathbb R^n$ satisfying $m = \Omega( \log(|S_1| \cdot |S_2|))$, for all $s_i \in S_i$, $|s_1^T E s_2| \leq C \sqrt{\frac{ \log(|S_1| \cdot |S_2|)}{m}} \cdot \|s_1\| \cdot \|s_2\|$.

The authors then provide guarantees for the optimizer in the context of these three settings, under these assumptions. More precisely, if $\overline v$ is the global optimizer, $\overline x$  is the true hidden signal, $x_G$ is the closest point in the range of the GAN to $\overline x$ and $\Delta = \overline \lambda_1 - \overline \lambda_2$ is the gap between the largest and second largest eigenvector of $\overline V$, they show,
$ \min \{\|\overline x - \overline v\|, \|\overline x + \overline v\| \} \leq O(\sqrt{k \log(Lr/\delta)/m}/\Delta) + 4(\overline \lambda_1/\Delta) \cdot \|x_G - \overline x\| + O(\delta n / m \Delta)$

Additionally, they demonstrate conditions on the initial $w_0$ which is input to PPower under which the algorithm has a linear rate of convergence, however they also need certain other assumptions which are satisfied in natural settings (such as the range of the GAN being positive).

Finally, they demonstrate experiments on MNIST for the spiked covariance model where they compare their algorithm (PPower) to the truncated power method (TPower) and the vanilla power method as a baseline. They show that the performance of PPower is much greater.





**Summary Of The Review:**

I think this is an interesting paper and vote to accept it, though it would be nice to see the experiments for the questions I've asked.

---

> ### Author Response · Authors · 2021-11-11
> **Responses to Reviewer 4eD5**
>
> Thanks for your recognition of this paper and the useful comments. Concerning the experiments for the questions you have asked, our responses are as follows:
>
> (**The Lipschitzness of GANs**) The Lipschitzness has become a standard assumption for theoretical studies (Bora et al., 2017, Dhar et al., 2018; Liu et al., 2020).  It is satisfied, for example, by any fully connected neural network with bounded weights and popular activation functions (such as the encoder and decoder of the VAE model used for the MNIST dataset), typically with $L$ on the order of $n^d$ when the depth is $d$. While this may seem large, it is alleviated by the mild $\log (L)$ dependence in the theory. We believe that checking the Lipschitz constant of a GAN is orthogonal to our main contributions, and best left to be investigated elsewhere.
>
> (**TPower in the wavelet basis**) For the MNIST dataset, the images are sparse in the natural basis, and we follow Bora et al., (2017) to perform the sparsity-based method TPower for these images in the natural basis directly. For the Fashion-MNIST dataset, indeed, some of the images are not sparse in the natural basis, but we observe from Figures 4 and 6 that even for the sparsest images (sandals), PPower also significantly outperforms TPower. We have noted this at the end of Appendix F in our revised version.

---

### Official Review · Reviewer_itua · 2021-11-08

**Correctness:** 4
**Technical Novelty And Significance:** 2
**Empirical Novelty And Significance:** 2
**Recommendation:** 5
**Confidence:** 4

**Main Review:**

The PCA problem under Generative priors studied here is a generalization of the standard PCA problem, and also a generalization of the cone-constrained PCA problem studied in [1].  I believe that studying PCA and the power iteration beyond convex settings (such as [1]) is well-motivated.  Since projection onto a general non-convex set is a computationally hard problem in general, this work assumes access to a black-box oracle that can project points to arbitrary non-convex sets.  Therefore, the main theoretical results of this work are of statistical nature: a bound on the sample complexity of the global maximizer of the constrained problem and a sample complexity analysis of power iteration (assuming that perfect projections to the non-convex set can be performed efficiently).

The first theoretical result (Theorem 1) (bounding the sample complexity of the global maximum) does not look very surprising to me: it uses the fact that the range of $L$-Lipschitz generative $G$ model has a small cover and relies on standard concentration/cover arguments.

 The convergence result of the power iteration (Theorem 2)  is more interesting as, even with a perfect projection oracle, the power iteration over general non-convex sets would not converge to the global maximum (constrained in the non-convex set).  The authors have some additional assumptions in order to prove convergence: namely that they initialize the iteration at some point that correlates well with the optimal solution $\bar{x}$, i.e., $\bar{x}^T w^{(0)} \geq \nu >0$.  The authors state that a similar assumption was used in [1] to prove convergence when the constrained set is a convex cone.  Under these assumptions it is not entirely clear to me whether the convergence proof presented here differs significantly from the prior work.  Given the above, I am leaning towards rejection of this work but I am willing to reconsider if the authors or other reviewers can show otherwise.

Feedback/Questions:
I think it would be beneficial if the authors compare their convergence analysis (Lemma 2, Theorem 2) with that of [1] (or other related prior work) and highlight the main differences/challenges of the non-convex setting (given the assumptions) analyzed here.

[1]: Yash Deshpande, Andrea Montanari, and Emile Richard. Cone-constrained principal component
analysis. In Conf. Neur. Inf. Proc. Sys. (NeurIPS), pp. 2717–2725, 2014.


**Summary Of The Paper:**

This work studies PCA under the contraint that the principal eigenvectors lie in the range of some fixed generative model, e.g., a neural network with fixed weights.  More precisely, given an $L$-Lipschitz function $G : \mathbb R^k \mapsto R(G)$ the authors try to find a vector $w \in R(G)$ that maximizes $w^T V w$, where $V = \bar{V} + E$, is a (noisy) version of the "true" psd matrix $\bar{V}$.  The corresponding true solution $\bar{x}$ is defined to be the top eigenvector of $\bar{V}$.

They assume that the range $R(G)$ of $G$ is a subset of the unit ball of $\mathbb R^n$ and that the error matrix $E$ is not too large in the sense that for any sets $S_1, S_2$ with $m = \Omega( \log( |S_1| |S_2|))$ it holds that $|s_1^T E s_2| \leq \sqrt{ \log(|S_1| |S_2|)/ m} \|s_1\|_2 \|s_2\|_2$.  This error bound is satisfied (via standard concentration inequalities) by the spiked covariance and phase retrieval model assuming that the number of samples $m$ is large enough.

The authors show two main results.  First, they show that under the above assumptions the global maximizer given the noisy matrix $V$, i.e., the maximizer $\hat{v}$ of $w^T V w$ constrained on $R(G)$ has error roughly $\|\hat{v} - \bar{x}\|_2 \leq \sqrt{k \log L/m}$.  Next, they show that the power iteration algorithm together with a projection step onto the contraint set $R(G)$ will also eventually converge to a vector with similar error.  They also give experimental results showing that adding this projection step in the power iteration leads to improved recovery for the spiked covariance and phase retrieval problems for the MNIST and Fashion-MNIST datasets in the case where the generative model $G$ is a pre-trained variational autoencoder of small latent dimension $k$.

**Summary Of The Review:**

Strengths:  The paper studies an important and well-motivated problem and provides sample complexity bounds for an interesting setting. Overall, the paper is well-written and the main results and contributions are cleanly stated.
Weaknesses: The first theoretical result does is not very surprising and its proof is rather standard.  It would also be good to discuss the technical novelty of the second theoretical result and compare it with the prior work.
Given the above, I am leaning towards rejection of this work but I am willing to reconsider if the authors or other reviewers can show otherwise.

---

> ### Author Response · Authors · 2021-11-11
> **Responses to Reviewer itua**
>
> Thanks for the helpful comments. Concerning the technical novelty of our Theorem 2, our responses are as follows.
>
> Indeed, both our Theorem 2 and [1,Theorem 3] (or [1,Corollary 4.1], which is simply a corollary of applying [1,Theorem 3] to the Gaussian noise model) require the initialization to have a positive scalar product with the underlying signal $\bar{\mathbf{x}}$. In fact, we expect that such a requirement is unavoidable without assuming that both $\bar{\mathbf{x}}$ and $-\bar{\mathbf{x}}$ are in the structured set of interest; see also the last two sentences of our Remark 4. However, making a similar assumption about the initialization by no means implies that our proof technique is also similar to that in [1].  We highlight the differences in the following:
>
> (1) Perhaps the most significant difference is that the proof of [1,Theorem 3] is highly dependent on the Moreau decomposition, which is only valid for a closed convex cone (see [1, Definition 1.2] in the supplementary material). In particular, the Moreau decomposition needs to be used at the beginning of the proof of [1,Theorem 3] in the supplementary material of [1], such as Eqs. (18) and (19). We do not see a way for the proof to proceed without the Moreau decomposition, and our $\mathrm{Range}(G)$ may be completely different from a convex cone.
>
> (2) We would like to highlight that one key observation in our proof of Lemma 2 (and thus Theorem 2) is that for a normalized generative model $G$, and any $\mathbf{x} \in \mathbb{R}^n$ and $a >0$, we have $\mathcal{P}_G(a \mathbf{x}) = \mathcal{P}_G(\mathbf{x})$ (Eq. (62)). This enables us to derive the important equation $\hat{\mathbf{s}} = \mathcal{P}_G(\mathbf{V} \mathbf{s}) = \mathcal{P}_G(\bar{\eta} \mathbf{V} \mathbf{s})$. We are not aware of a similar equation in the proof of [1,Theorem 3].
>
> (3) In the PPower method in [1], the authors need to add $\rho \mathbf{I}_n$ with $\rho >0$ to the observed data matrix $\mathbf{V}$ to improve the convergence. In particular, they mention in the paragraph before the statement of Theorem 3 that “the memory term $\rho \mathbf{v}^{t}$ is necessary for our proof technique to go through”. In contrast, our proof of Theorem 2 does not require adding such terms, even when our data model is restricted to the spiked Wigner model considered in [1].
>
> (4) Last but not least, we want to emphasize that our problem formulation is novel and we consider a matrix model that is significantly more general than the spiked Wigner model studied in [1].
>
> We have added a modified version of the above discussion into Appendix E in our revised paper.
>
> [1]: Yash Deshpande, Andrea Montanari, and Emile Richard. Cone-constrained principal component analysis. In Conf. Neur. Inf. Proc. Sys. (NeurIPS), pp. 2717–2725, 2014.

---

### Author Response · Authors · 2021-11-11
**Thanks for the three anonymous reviewers**

We are very grateful to the reviewers for their helpful feedback and suggestions. Our responses to the main concerns are given to each reviewer separately, and we have made the corresponding revisions (in blue) in our revised version.

---

### Public Comment · ~Zhaoqiang_Liu1 · 2022-09-08
**Updated experiments in arXiv version**

Dear Readers,

After further investigation of the experiments in this ICLR paper, we have decided to upload an update to arXiv (https://arxiv.org/abs/2203.09693) to address a couple of issues, as well as update the code available on GitHub.  Note that our main contributions are all theoretical, and the theory sections remain unchanged, but we still believe that these experimental changes ought to be highlighted.

In particular, we note the following:
1) When computing Cosine Similarity, we mistakenly omitted the absolute value operation, which is required because any signal and its negative should be treated as equivalent in our problem.  The performance of some baselines slightly improved after correcting this, but the overall conclusions remain the same.
2) We regretfully found our Fashion-MNIST experiments (Appendix H) to have reproducibility issues, and we have now updated them to correct this. Notably, we switched the generative model from BEGAN to VAE, which we found to be most consistent in this sense. The pre-trained model can be downloaded from https://drive.google.com/file/d/1tXIRAbdtDrip_8Wu0-LbTvr2aGHSjx9R/view?usp=sharing (it is a bit too large (~160Mb) to be uploaded into the GitHub repository https://github.com/liuzq09/GenerativePCA directly).

We apologize for the inconvenience and for not detecting these issues prior to submission.

Finally, regarding the theoretical results, we note that the arXiv submission additionally includes an algorithm-independent lower bound (Appendix J therein) which was not included in the ICLR paper.

Regards,
Authors (Liu/Liu/Ghosh/Han/Scarlett)

---

### Decision · Program_Chairs · 2022-01-20

**Decision:**

Accept (Poster)

**Comment:**

This paper studies PCA under a generative model setup. The authors analyze the projected power method in a range of natural settings.
Moreover, experimental evaluation and comparison to other methods is performed on MNIST. The paper studies an important problem. Despite some initial concerns, the reviewers overall agreed that this is an interesting contribution. I recommend acceptance.